# TLR7 mediated viral recognition results in focal type I interferon secretion by dendritic cells

Shin-Ichiroh Saitoh[1], Fumiko Abe[2], Atsuo Kanno[1], Natsuko Tanimura[1], Yoshiko Mori Saitoh[1], Ryutaro Fukui[1], Takuma Shibata[1], Katsuaki Sato[3], Takeshi Ichinohe[4], Mayumi Hayashi[5], Kazuishi Kubota [5], Hiroko Kozuka-Hata[6], Masaaki Oyama[6], Yorifumi Kikko[2], Toshiaki Katada[2], Kenji Kontani[2,7] & Kensuke Miyake[1,8]

Plasmacytoid dendritic cells (pDC) sense viral RNA through toll-like receptor 7 (TLR7), form self-adhesive pDC–pDC clusters, and produce type I interferons. This cell adhesion enhances type I interferon production, but little is known about the underlying mechanisms. Here we show that MyD88-dependent TLR7 signaling activates CD11a/CD18 integrin to induce microtubule elongation. TLR7$^+$ lysosomes then become linked with these microtubules through the GTPase Arl8b and its effector SKIP/Plekhm2, resulting in perinuclear to peripheral relocalization of TLR7. The type I interferon signaling molecules TRAF3, IKKα, and mTORC1 are constitutively associated in pDCs. TLR7 localizes to mTORC1 and induces association of TRAF3 with the upstream molecule TRAF6. Finally, type I interferons are secreted in the vicinity of cell–cell contacts between clustered pDCs. These results suggest that TLR7 needs to move to the cell periphery to induce robust type I interferon responses in pDCs.

[1] Division of Innate Immunity, Department of Microbiology and Immunology, The Institute of Medical Science, The University of Tokyo, 4-6-1 Shirokanedai, Minato-ku, Tokyo 108-8639, Japan. [2] Department of Physiological Chemistry, Graduate School of Pharmaceutical Sciences, The University of Tokyo, Bunkyo-ku, Tokyo 113-0033, Japan. [3] Division of Immunology, Department of Infectious Diseases, Faculty of Medicine, University of Miyazaki, 5200 Kihara, Kiyotake, Miyazaki 889-1692, Japan. [4] Division of Viral Infection, Department of Infectious Disease Control, International Research Center for Infectious Diseases, Institute of Medical Science, The University of Tokyo, Minato-ku, Tokyo 108-8639, Japan. [5] Discovery Science and Technology Department, Daiichi Sankyo RD Novare Co., Ltd., 1-16-13 Kitakasai, Edogawa-ku, Tokyo 134-8630, Japan. [6] Medical Proteomics Laboratory, The Institute of Medical Science, The University of Tokyo, 4-6-1 Shirokanedai, Minato-ku, Tokyo 108-8639, Japan. [7] Department of Biochemistry, Meiji Pharmaceutical University, Kiyose, Tokyo 204-8588, Japan. [8] Laboratory of Innate Immunity, Center for Experimental Medicine and Systems Biology, The Institute of Medical Science, The University of Tokyo, Minato-ku, Tokyo 108-8639, Japan. Shin-Ichiroh Saitoh, Kenji Kontani and Kensuke Miyake contributed equally to this work. Correspondence and requests for materials should be addressed to K.M. (email: kmiyake@ims.u-tokyo.ac.jp)

T he innate immune system is the first line of defense against microbial infection and plasmacytoid dendritic cells (pDC) are one of the most important innate immune cells[1, 2]. In response to virus infection, pDCs produce type I interferons that are critically required for host protection against viruses. Toll-like receptors (TLR) are pattern recognition receptors that can bind microbial products to activate immune responses[3]. TLR7 and TLR9 are expressed by pDCs and sense viral RNA and DNA, respectively.

Viral infection stimulates type I interferon-dependent clustering of pDCs[4]. On the other hand, pDC adhesion enhances expression of type I interferons[5, 6] and type I interferon

production by pDCs is positively correlated with cell density[6]. These results indicate the existence of a positive feedback loop between type I interferons and cell adhesion. Type I interferons are likely to activate cell adhesion molecules, and cell–cell contact, in turn, may enhance type I interferon signaling. Autocrine/paracrine type I interferon signaling is important for full activation of pDCs[4, 7, 8]. However, the role of cell adhesion in type I interferon signaling has not been clarified.

pDC TLR7 and TLR9 induce proinflammatory cytokines and IFN-α/β. IFN-α/β are distinct from proinflammatory cytokines in an additional requirement for signaling molecules; whereas MyD88, interleukin-1 receptor-associated kinase 4 (IRAK4), and

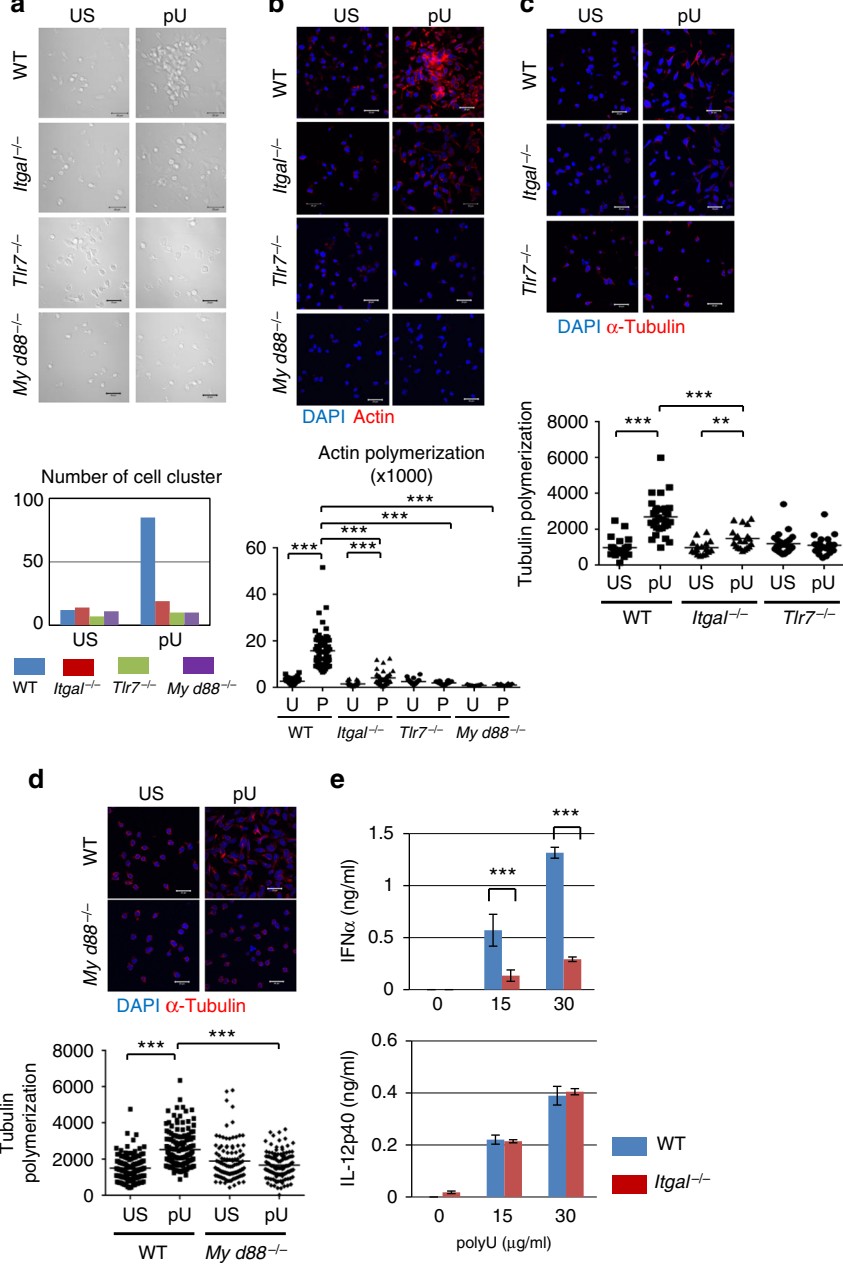

**Fig. 1** Impaired pDC clustering and IFN-α production in *Itgal*⁻/⁻ pDCs. **a** WT, *Itgal*⁻/⁻, *Tlr7*⁻/⁻, and *Myd88*⁻/⁻ BM-pDCs were left unstimulated (US) or activated (pU) with polyU at 25 µg/mL for 4 h. Clustering of pDCs was visualized by microscopy and counted in 40 visual areas. Scale bar, 20 µm. **b** Actin was stained and fluorescence intensity was measured for statistical analyses (*n* > 23). **c**, **d** α-tubulin was stained and fluorescence intensities of α-tubulin staining in each cell were measured for statistical analysis (*n* > 16). **e** WT and *Itgal*⁻/⁻ BM-pDCs were stimulated or not with polyU at 15 and 30 µg/mL for 24 h. Production of IFN-α and IL-12 p40 was measured by ELISA. Data shown are mean ± s.d. from triplicate well. The experiments were repeated three times with similar results. ***P < 0.001, **P < 0.01 (unpaired two-tailed *t*-test)

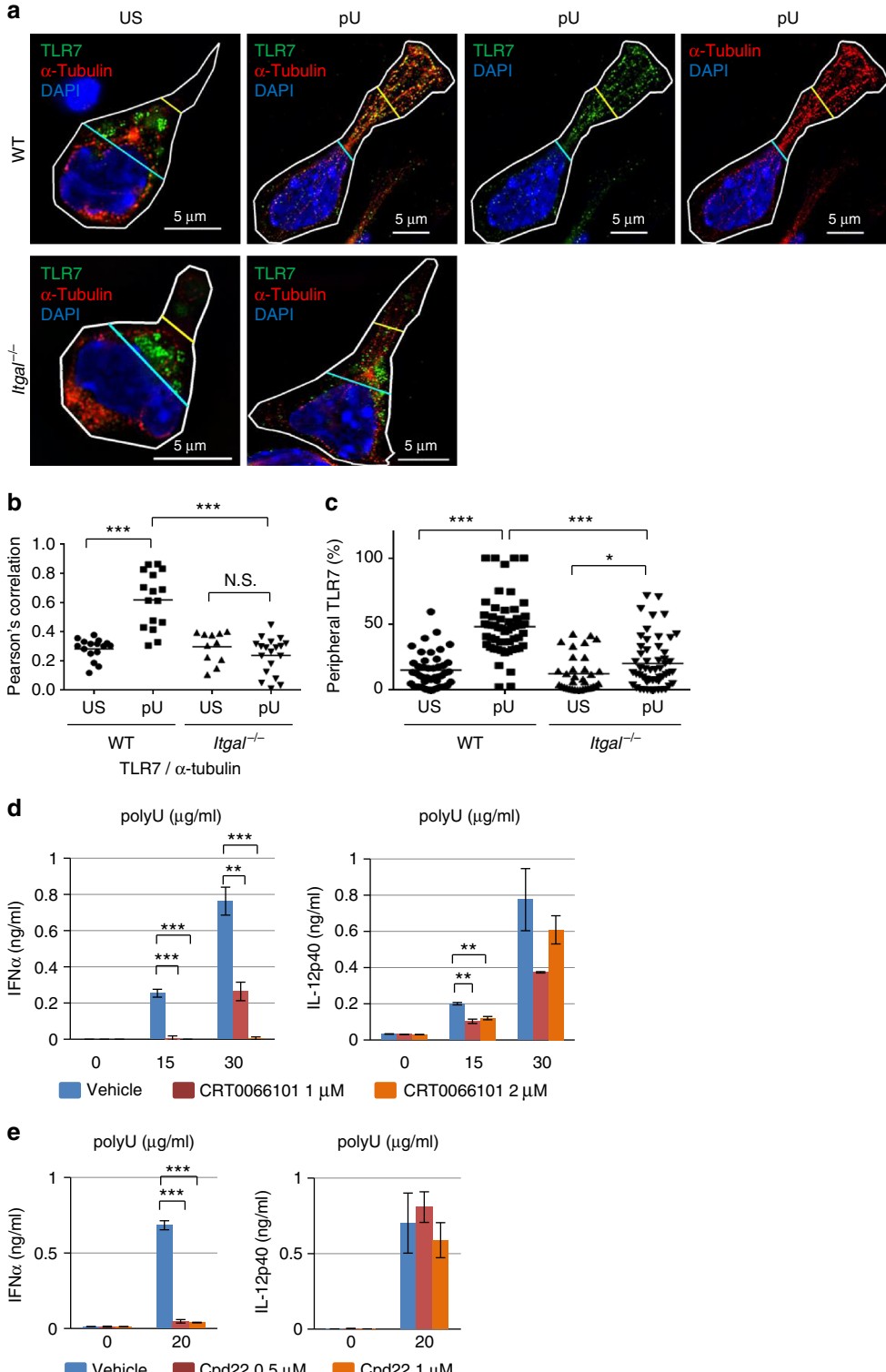

**Fig. 2** TLR7 trafficking to cell periphery and IFN-α induction depend on CD11a/CD18 activation. **a** WT and *Itgal*[−/−] BM-pDCs were either left unstimulated (US) or activated (pU) with 25 μg/mL polyU for 4 h. Abs against TLR7 (green), and α-tubulin (red) were used together with DAPI staining of cell nuclei (blue) prior to analysis by structured illumination microscopy (SIM). Plasma membranes are indicated by white line. Blue lines are placed at the boundary between cytoplasm and nucleus. Yellow lines are at the middle of blue lines and the tip of the polarized cytoplasm. These lines were used for quantification of TLR7 trafficking. **b** Quantification of TLR7 co-localization with α-tubulin (*n* > 10). **c** To quantify TLR7 trafficking to the cell periphery, the cytoplasm was first divided by perinuclear and peripheral regions as indicated by blue and yellow lines (**a**). Shown is a percentage of peripheral TLR7 (*n* > 33). **d**, **e** BM-pDCs were pretreated with protein kinase D (PKD) inhibitor CRT0066101 (**d**) or integrin-linked kinase (ILK) inhibitor Cpd22 (**e**) for 1 h at the indicated concentration, and then stimulated with polyU at 15, 30, or 20 μg/mL for 24 h. Production of IFN-α and IL-12 p40 was measured by ELISA. Data shown are mean ± s.d. from triplicate wells. The experiments were repeated three times with similar results. 5 μm (**a**). ***$P < 0.001$, **$P < 0.01$, *$P < 0.05$ (unpaired two-tailed *t*-test)

TNF receptor-associated factor 6 (TRAF6) are required for both cytokine families, IRAK1, TRAF3, IKKα, and interferon regulatory factor 7 (IRF7) are additionally required for IFN-α induction[9–12]. Ligand-dependent trafficking of TLR7 and TLR9 is another unique requirement for IFN-α/β induction in pDCs. TLR7 and TLR9 are distributed in endosomes and lysosomes, and upon activation adaptor protein 3 (AP3)-dependently move from endosomes to lysosomes and this trafficking is required for induction of IFN-α/β, but not for proinflammatory cytokines[13–15]. AP3 facilitates interaction between signaling molecules TRAF3 and IRF7[13]; however, why IFN-α/β induction requires TLR7/9 trafficking is not known.

Here we examine the role of TLR7 trafficking in IFN-α induction by TLR7 and show that cell adhesion is required for TLR7 trafficking. pDCs lacking CD11a/CD18 integrin have decreased IFN-α expression in response to TLR7 activation as a result of impaired TLR7 trafficking. Activation of CD11a/CD18 integrin by TLR7 induces microtubule polymerization. Lysosomal TLR7 is linked with microtubules through the lysosomal GTPase Arl8b and its effector SifA and kinesin-interacting protein (SKIP; also known as Plekhm2), leading to anterograde TLR7 trafficking. TLR7 trafficking enables ligand-dependent interaction of the two downstream signaling molecules for IFN-α induction, TRAF3, and TRAF6. TRAF3 is steadily associated with downstream molecules IKKα and phosphorylates mammalian target of rapamycin complex 1 (mTORC1). These results show that TLR7 trafficking is the molecular mechanism to account for type I interferon control by cell–cell adhesion of pDCs.

## Results

**CD11a/CD18 integrin is required for pDC IFN-α production**. The present study addressed the role of cell adhesion in IFNα/β induction. Cell adhesion via CD11a/CD18 integrin enhances IFN-α/β production by pDCs[5, 16]. CD11a/CD18 and its ligands, CD54 (also known as ICAM-1) and CD102 (ICAM-2), were expressed on pDCs (Supplementary Fig. 1a). To study the role of CD11a/CD18 integrin in TLR7-induced antiviral responses, wild-type (WT) pDCs or those deficient in TLR7 ($Tlr7^{-/-}$), CD11a ($Itgal^{-/-}$), or MyD88 ($Myd88^{-/-}$) were stimulated with polyU single-stranded RNA. We found that polyU exposure induced pronounced clustering and polymerization of actin and microtubule in WT BM-pDCs but not their $Tlr7^{-/-}$, $Itgal^{-/-}$, or $Myd88^{-/-}$ counterparts (Fig. 1a–d). This indicated that TLR7 activates CD11a/CD18-mediated cell adhesion through the Myd88-dependent signaling pathway.

In addition to cluster formation and cytoskeletal changes, IFN-α production was also significantly impaired in $Itgal^{-/-}$ pDCs, whereas IL-12p40 expression was unaltered (Fig. 1e). This was the case irrespective of whether the cells were stimulated with polyU or the alternative TLR7 ligand loxoribine (Supplementary Fig. 1b). The expression of TLR7, CD54, CD102, and signaling molecules downstream of TLR7 in $Itgal^{-/-}$ and $Myd88^{-/-}$ BM-pDCs were not altered (Supplementary Fig. 1a, c, d–g). To further confirm the requirement of CD11a/CD18 for IFN-α expression, we used a CD11a/CD18 inhibitor, RWJ50271, which significantly inhibited pDC clustering, microtubule polymerization, and IFN-α expression, but not IL-12 p40 expression in polyU-activated pDCs (Supplementary Fig. 2a–c). Together, these data indicate that TLR7-mediated activation of integrin CD11a/CD18 is required for IFN-α induction and suggest that pDC clusters are the critical sites of IFN-α expression.

**TLR7 trafficking are impaired in $Itgal^{-/-}$ pDCs**. TLR7 is localized in endosomes/lysosomes and ligand-activated trafficking of TLR7 is essential for induction of IFN-α/β[13, 14, 17]. The changes in

adhesive properties could potentially alter IFN-α/β induction by impacting TLR7 distribution within these cells. To address this possibility, we visualized TLR7 using immunostaining and analysis via structured illumination microscopy (SIM). Both before and after polyU stimulation, TLR7 was observed to co-localize with the lysosomal marker LAMP-2 (Supplementary Fig. 3a, b) but not with early endosomal marker Rab5 (Supplementary Fig. 3b, c), indicating that TLR7 is selectively localized to LAMP-2+ lysosomes. We next assessed whether this selective distribution of ligand-activated TLR7 depended on microtubules, which are thought to be essential for lysosome trafficking. Exposure to TLR7 ligand polyU induced microtubule polymerization and extension from the perinuclear region to the cell periphery in WT pDCs, but this process was significantly impaired in $Tlr7^{-/-}$ and $Itgal^{-/-}$ pDCs (Figs. 1c, 2a). TLR7 displayed significantly higher co-localization with microtubules after ligand activation, and polyU exposure increased the proportion of TLR7 localized to the cell periphery by >30% in WT pDCs (Fig. 2b, c). The co-localization between TLR7 and LAMP-2 did not change between perinuclear and peripheral regions (Supplementary Fig. 3d), suggesting that TLR7-containing LAMP-2+ lysosomes translocate from perinuclear to peripheral regions.

$Itgal^{-/-}$ pDCs exhibited only weak microtubule polymerization compared with WT pDCs (Fig. 1c), while ligand-activated TLR7 displayed reduced co-localization with microtubules and failed to traffic to the cell periphery (Fig. 2a–c). RWJ50271, the LFA-1 inhibitor, also inhibited TLR7 trafficking to the cell periphery (Supplementary Fig. 2d). These results suggest that ligand-activated TLR7 stimulates microtubule polymerization and elongation in a CD11a/CD18-dependent manner to allow translocation of TLR7-containing lysosomes from a perinuclear location to the cell periphery.

We analyzed the inside-out signaling to activate CD11a/CD18 integrin. Protein kinase C (PKC) family is critical for inside-out integrin activation in T cell[18, 19]. We used a PKC inhibitor, GO6983, and found that GO6983 significantly inhibited IFN-α expression (Supplementary Fig. 4a). Next, we analyzed one of the PKC downstream molecule, protein kinase D1 (PKD1)[20]. Consistent with a previous report using macrophages[21], PKD1 was phosphorylated upon TLR7 activation in WT pDCs but not in $Myd88^{-/-}$ pDCs (Supplementary Fig. 4b, c). Furthermore, we confirmed that a PKD inhibitor, CRT0066101, suppressed IFN-α expression, but much less IL-12p40 (Fig. 2d). PKD1 associates with and activates Ras-related protein 1 (Rap1), a well-known CD11a/CD18 activator[22, 23]. We therefore examined TLR7-dependent activation of Rap1 as judged by its binding to RalGDS Rap-binding protein. Rap1 pull-down with RalGDS Rap-binding domain was detected 4 h after polyU stimulation (Supplementary Fig. 4d). These results suggest that the TLR7–MyD88–PKC–PKD–Rap1 axis activates CD11a/CD18 integrin.

The signaling pathway from CD11a/CD18 to microtubule polymerization was next studied. A previous report shows that integrin-linked kinase (ILK) is required for polarization of microtubule and lytic granule by β2 integrin in natural killer (NK) cells[24]. Therefore, pDCs were treated with an ILK inhibitor Cpd22 and cytokine production was examined. IFN-α production was completely abolished, while IL-12 p40 production was unaltered (Fig. 2e). These findings suggest that TLR7 activates the integrin CD11a/CD18 to induce microtubule polymerization, which is needed for TLR7 trafficking to the peripheral regions of pDCs and subsequent IFN-α induction.

**TLR7 is associated with Arl8b**. TLR7 co-localization with microtubules was significantly increased after polyU stimulation

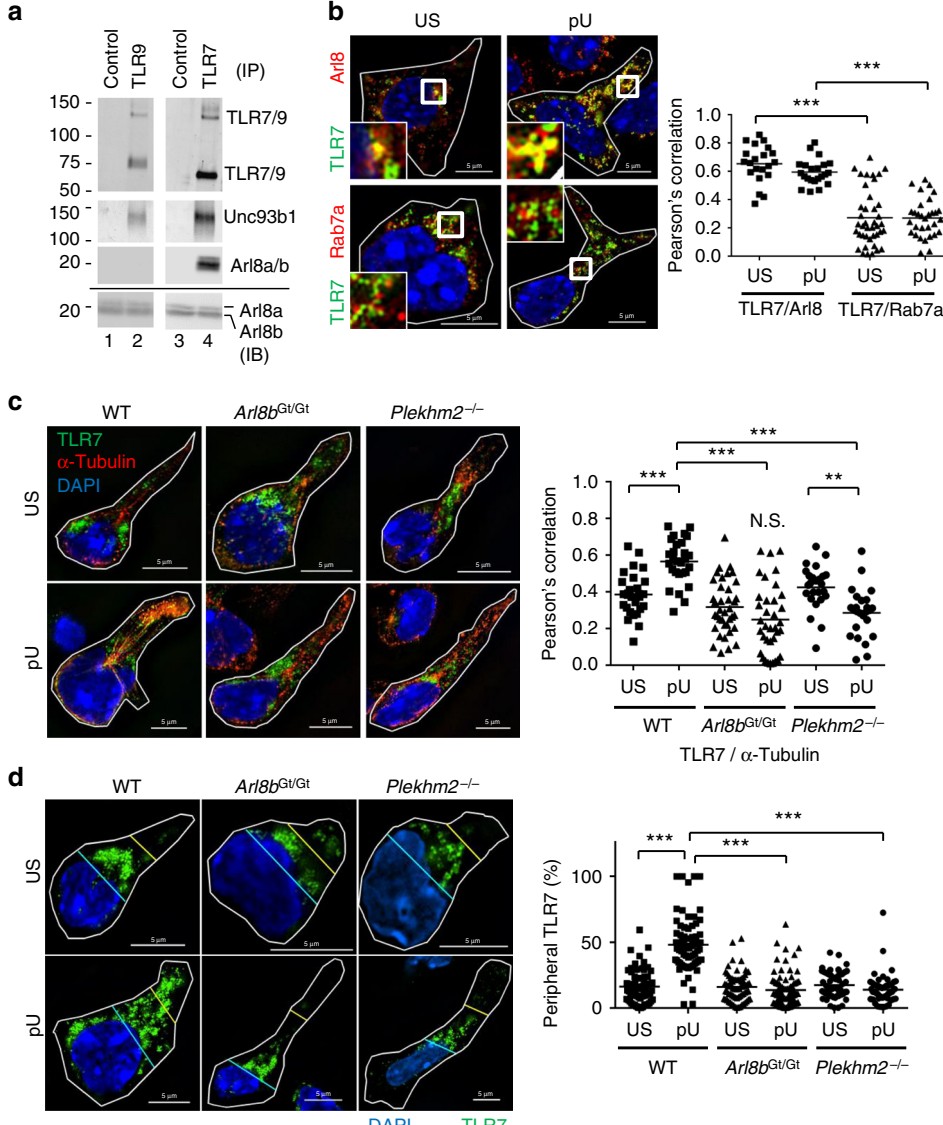

**Fig. 3** Impaired TLR7 trafficking in *Arl8b*[Gt/Gt] and *Plekhm2*[−/−] pDCs. **a** TLR7 and TLR9 were immunoprecipitated from BM-pDCs and subjected to immunostaining of TLR7, TLR9, Unc93b1, and Arl8a/b. Lanes 1 and 3 show immunoprecipitations with isotype-matched control Ab. The lowermost blot shows immunostaining of whole cell lysate with Ab against Arl8a/b. Apparent molecular mass is indicated (left). **b** BM-pDCs were either left unstimulated (US) or activated (pU) with 25 µg/mL polyU for 3 h prior to staining with Ab against TLR7 (green), Arl8a/b (red, upper panels), and Rab7a (red, lower panels). Nuclei were visualized by DAPI staining (blue). Higher magnification images of boxed regions are shown in the insets. Also shown is the statistical analysis of TLR7 co-localization with Arl8 or Rab7a (n > 20). **c** WT, *Arl8b*[Gt/Gt], and *Plekhm2*[−/−] BM-pDCs were either left unstimulated (US) or activated with 25 µg/mL polyU (pU) for 4 h prior to Ab staining of TLR7 (green), α-tubulin (red), and DAPI staining of cell nuclei (blue). Also shown is the statistical analysis of TLR7 co-localization with microtubule (n > 24). **d** WT, *Arl8b*[Gt/Gt], and *Plekhm2*[−/−] BM-pDCs were either left unstimulated (US) or activated with 25 µg/mL polyU (pU) for 3 h prior to Ab staining of TLR7 and DAPI staining. Right panel (**d**) shows the percentages of peripheral TLR7 (n > 53). WT samples shown here are in part from Fig. 2c. ***P < 0.001, **P < 0.01 (unpaired two-tailed *t*-test)

(Fig. 2a, b), so we next examined the molecular machinery linking TLR7-containing lysosomes to polymerized microtubules. To do this, we used mass spectrometry to identify molecules associated with the multiple transmembrane protein Unc93B1, which regulates endosomal TLR distribution[25, 26]. Using this approach, we consistently detected close association of Unc93B1 with the small GTPases Arl8a and Arl8b (Supplementary Fig. 5a–c). Accordingly, immunoprecipitation of Unc93B1 in a macrophage cell line transduced to overexpress this protein resulted in co-precipitation of Arl8a and Arl8b but not the alternative GTPase Rab5 (Supplementary Fig. 5d). Furthermore, when we used the same approach to precipitate TLR7 from total spleen cells, cDCs, or pDCs, the receptor co-precipitated with Arl8a and Arl8b as well

as Unc93B1 (Fig. 3a; Supplementary Fig. 5e). In contrast, TLR9 hardly co-precipitated with Arl8a and Arl8b. When we analyzed the subcellular distribution of TLR7, we detected marked co-localization with Arl8a and/or Arl8b which was not altered upon polyU stimulation (Fig. 3b). Both Rab7a and Arl8b have been reported to control lysosomal movement[27], but TLR7 displayed far greater co-localization with Arl8a/b than with Rab7a, suggesting that TLR7 is preferentially sequestered into Arl8[+] lysosomes rather than Rab7a[+] late endosomes.

**Arl8b is required for TLR7 trafficking and IFN-α expression.** Since Arl8a and Arl8b are highly homologous proteins that

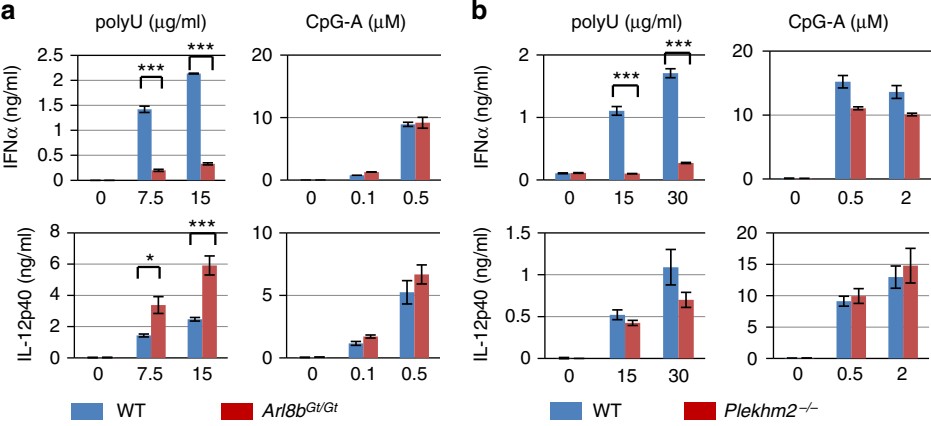

**Fig. 4** Arl8b and Plekhm2 are essential for IFN-α expression. **a, b** WT (blue), *Arl8b*^Gt/Gt (red), and *Plekhm2*^−/− (red) BM-pDCs were stimulated or not with either polyU or CpG-A at the indicated concentrations. Production of IFN-α and IL-12p40 was measured by ELISA. Data are presented as mean ± s.d. of triplicate wells. Similar experiments were repeated three or four times. ***$P < 0.001$, *$P < 0.05$ (unpaired two-tailed *t*-test)

exhibit 91% amino-acid identity (Supplementary Fig. 5c), we focused subsequent analyses on Arl8b, which has a well-documented role in anterograde lysosomal trafficking[27–29]. To explore the importance of Arl8b in regulating TLR7 responses, we obtained *Arl8b*^Gt/Gt gene trap mice (Supplementary Fig. 6a) and assessed receptor distribution and function in pDCs. While a considerable number of *Arl8b*^Gt/Gt mice died from unknown causes, the rest of animals developed normally (Supplementary Fig. 6b, c). Arl8b messenger RNA (mRNA) and protein were not detectable in *Arl8b*^Gt/Gt pDCs, whereas the levels of Arl8a mRNA and protein were unaltered (Supplementary Fig. 6d, e). The expression levels of TLR7 mRNA and protein were also comparable with WT controls (Supplementary Fig. 6f–h). We then assessed the impact of *Arl8b* deficiency on polyU-activated microtubule polymerization and TLR7 trafficking in pDCs. PolyU-induced pDC clustering was comparably observed in WT and *Arl8b*^Gt/Gt cells (Supplementary Fig. 7a). Also, polyU induced microtubule polymerization in *Arl8b*^Gt/Gt BM-pDCs (Supplementary Fig. 7b). However, ligand-activated TLR7 failed to co-localize with microtubules and to traffic to the cell periphery in *Arl8b*^Gt/Gt BM-pDCs (Fig. 3c, d). These results suggest that Arl8b is required for the interaction between TLR7-containing lysosomes and microtubule but not receptor-mediated activation of integrin CD11a/CD18.

Arl8b binds to SifA and kinesin-interacting protein SKIP (also known as Plekhm2), which contributes to the regulation of anterograde lysosomal trafficking[30]. We therefore investigated whether SKIP influences TLR7 trafficking in pDCs by assessing receptor distribution in *Plekhm2*^−/− pDCs, in which the lack of *Plekhm2* mRNA was verified by PCR (Supplementary Fig. 8a). Whereas polyU-activated microtubule polymerization and co-localization between LAMP-2 and TLR7 were not impaired (Supplementary Figs. 7b, 8b), the co-localization between TLR7 and microtubule and TLR7 trafficking to cell periphery were impaired in *Plekhm2*^−/− BM-pDCs (Fig. 3c, d). The expression of TLR7, CD11a/CD18, and TLR7 signaling molecules was not altered in *Plekhm2*^−/− BM-pDCs (Supplementary Fig. 8c–e). As expected, *Plekhm2*^−/− pDCs resemble *Arl8b*^Gt/Gt pDCs, indicating that the Arl8b-SKIP axis links TLR7-containing lysosomes to polymerized microtubules for the transport to the cell periphery.

Upon stimulation with polyU or loxoribine, *Arl8b*^Gt/Gt pDCs displayed significant impairment of IFN-α induction (Fig. 4a; Supplementary Fig. 8f), whereas expression levels of proinflammatory cytokine IL-12p40 and TLR9-mediated responses to CpG-A were unimpaired. In fact, IL-12p40 production was significantly enhanced in *Arl8b*^Gt/Gt pDCs for a currently

unknown reason. *Plekhm2*^−/− pDCs also displayed a selective defect in TLR7-dependent IFN-α production comparable to that observed in their *Arl8b*^Gt/Gt counterparts (Fig. 4b; Supplementary Fig. 8g). These results demonstrated that the anterograde lysosomal movement under the control of Arl8b and SKIP is required for IFN-α induction by TLR7. Taken together with our earlier observation that TLR7 trafficking is disrupted in *Itgal*^−/− pDCs (Fig. 2c), these findings indicated that TLR7 activates cell surface integrin CD11a/CD18 to induce microtubule polymerization, and the lysosomal Arl8b-SKIP axis to link TLR7 with extended microtubules. TLR7 trafficking to the cell periphery ultimately enabled IFN-α induction.

The phenotypes found in both *Arl8b*^Gt/Gt and *Plekhm2*^−/− BM-pDCs reminded us of *AP3b1*^−/− BM-pDCs, in which TLR7-dependent expression of IFN-α/β, but not inflammatory cytokines, was impaired[13]. Ligand-dependent TLR9 trafficking to lysosomes requires adaptor protein 3 (AP3)-dependent endosomal trafficking[13]. Arl8b-dependent lysosomal trafficking studied here may also depend on AP3. The co-localization between Arl8 and AP3 δ, a subunit of AP3, was analyzed in WT BM-pDCs. We could not see high co-localization between Arl8 and AP3 δ in steady-state BM-pDCs. Unexpectedly, polyU stimulation decreased the co-localization between Arl8 and AP3 δ (Supplementary Fig. 9). These results showed sharp contrast to the previous finding that co-localization between TLR9 and AP3 increased by ligand stimulation[13].

**Arl8b is required for IFN-α production against influenza virus.** We next investigated the role of Arl8b in TLR7 responses to influenza virus infection of pDCs. Influenza virus induced pronounced clustering of WT pDCs and *Arl8b*^Gt/Gt pDCs, but not *Tlr7*^−/− pDCs (Fig. 5a). While virus exposure stimulated marked expression of IL-12p40 in *Arl8b*^Gt/Gt pDCs, these cells failed to produce IFN-α under the same conditions (Fig. 5b). Influenza virus induced TLR7 co-localization with microtubules and extended these to the cell periphery in WT pDCs but not *Arl8b*^Gt/Gt pDCs (Fig. 5c). Consequently, TLR7 was localized to the cell periphery in WT pDCs, but much less in *Arl8b*^Gt/Gt pDCs (Fig. 5d). Despite exhibiting normal virus-induced clustering behavior, microtubule elongation was impaired in *Arl8b*^Gt/Gt pDCs. Although microtubule polymerization was detected in polyU-activated *Arl8b*^Gt/Gt pDCs (Supplementary Fig. 7b), we cannot exclude a possibility that Arl8b contributes partially to microtubule elongation particularly in influenza infection. These

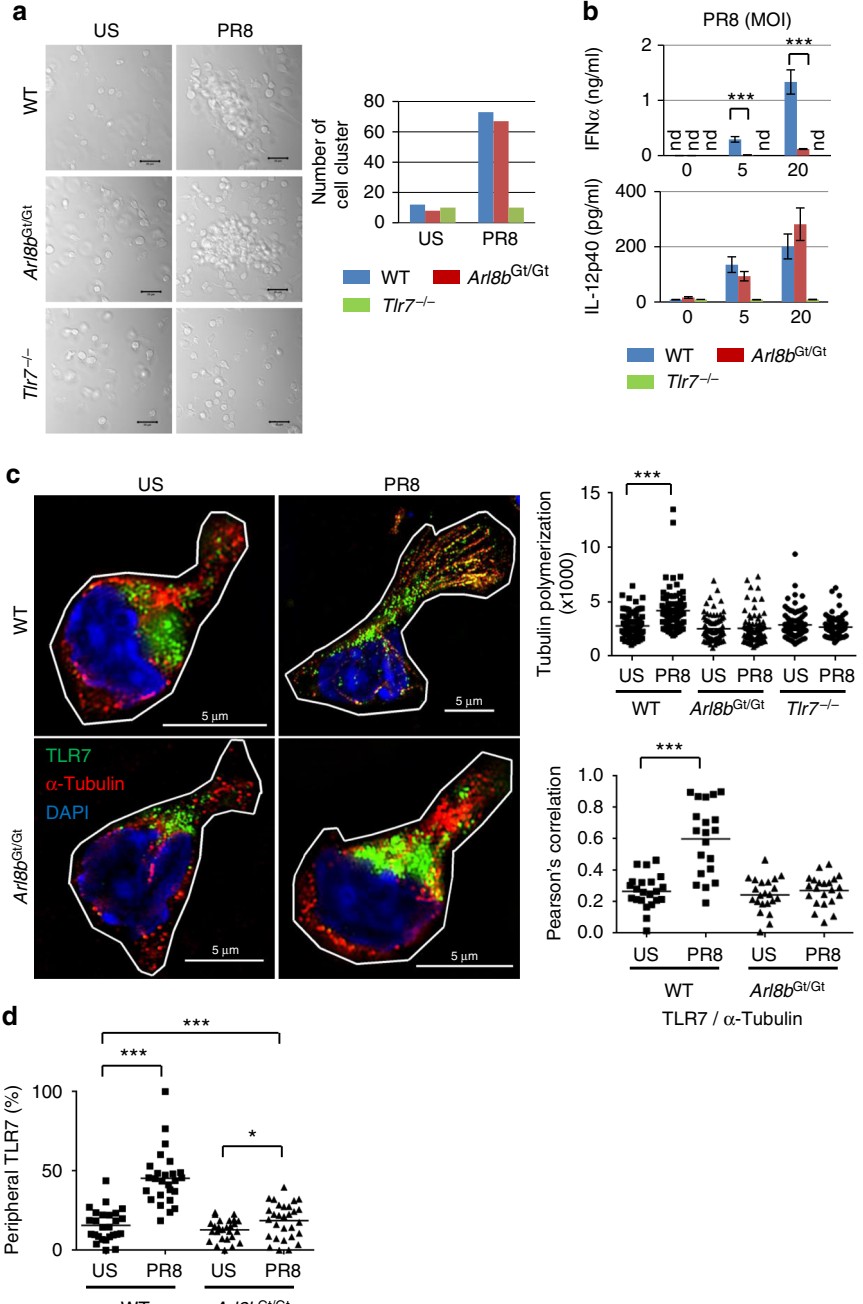

**Fig. 5** Impaired TLR7 trafficking and defective IFN-α responses in influenza virus-infected *Arl8b*^Gt/Gt pDCs. WT, *Arl8b*^Gt/Gt, or *Tlr7*^−/− pDCs were left unstimulated (US) or infected with influenza virus PR8 (PR8) at 20 MOI (**a**, **c**) or at the indicated doses (**b**). After 5 h, pDC clustering was assessed by confocal microscopy and enumerated in 30 visual areas (**a**). The cells were immunolabeled using Abs against TLR7 (green), and α-tubulin (red), together with DAPI staining of cell nuclei (blue). Statistical determination of microtubule polymerization ($n > 84$), TLR7 co-localization with α-tubulin ($n > 19$), and TLR7 trafficking in each cell are also shown ($n > 25$) (**c**, **d**). After 24 h, cytokine production was determined by ELISA (**b**). Data are expressed as mean ± s.d. from triplicate wells. Similar experiments were repeated three times. Scale bar, 20 μm (**a**), 5 μm (**c**). ***$P < 0.001$ (unpaired two-tailed *t*-test)

results demonstrated that, as with polyU exposure, Arl8b-dependent TLR7 trafficking in pDCs is required for IFN-α responses to live influenza virus.

**TLR7 traffics to activate IFN-α-inducing signaling molecules.** The importance of TLR7 trafficking in the IFN-α signaling pathway was next addressed. TRAF6 is an E3 ubiquitin ligase that is recruited to TLRs and mediates activation of NF-κB and IRF7[10, 31]. When we immunoprecipitated the TRAF6 signaling

molecule, we detected co-precipitation of TRAF3 and IRF7 after polyU stimulation of WT pDCs but not *Arl8b*^Gt/Gt pDCs (Fig. 6a). IKKα is a signaling molecule working between TRAF3 and IRF7[10–12]. Furthermore, IFN-α/β induction in pDCs requires mTORC1[32, 33]. To study the relationship among these signaling molecules, we immunoprecipitated TRAF3 from pDCs, and observed marked co-precipitation of IKKα, phosphorylated mTOR (p-mTOR), and RAPTOR a component of mTORC1 (Fig. 6b). Co-precipitation of IKKα and mTORC1 with TRAF3 was detectable in unstimulated cells as well as from *Arl8b*^Gt/Gt

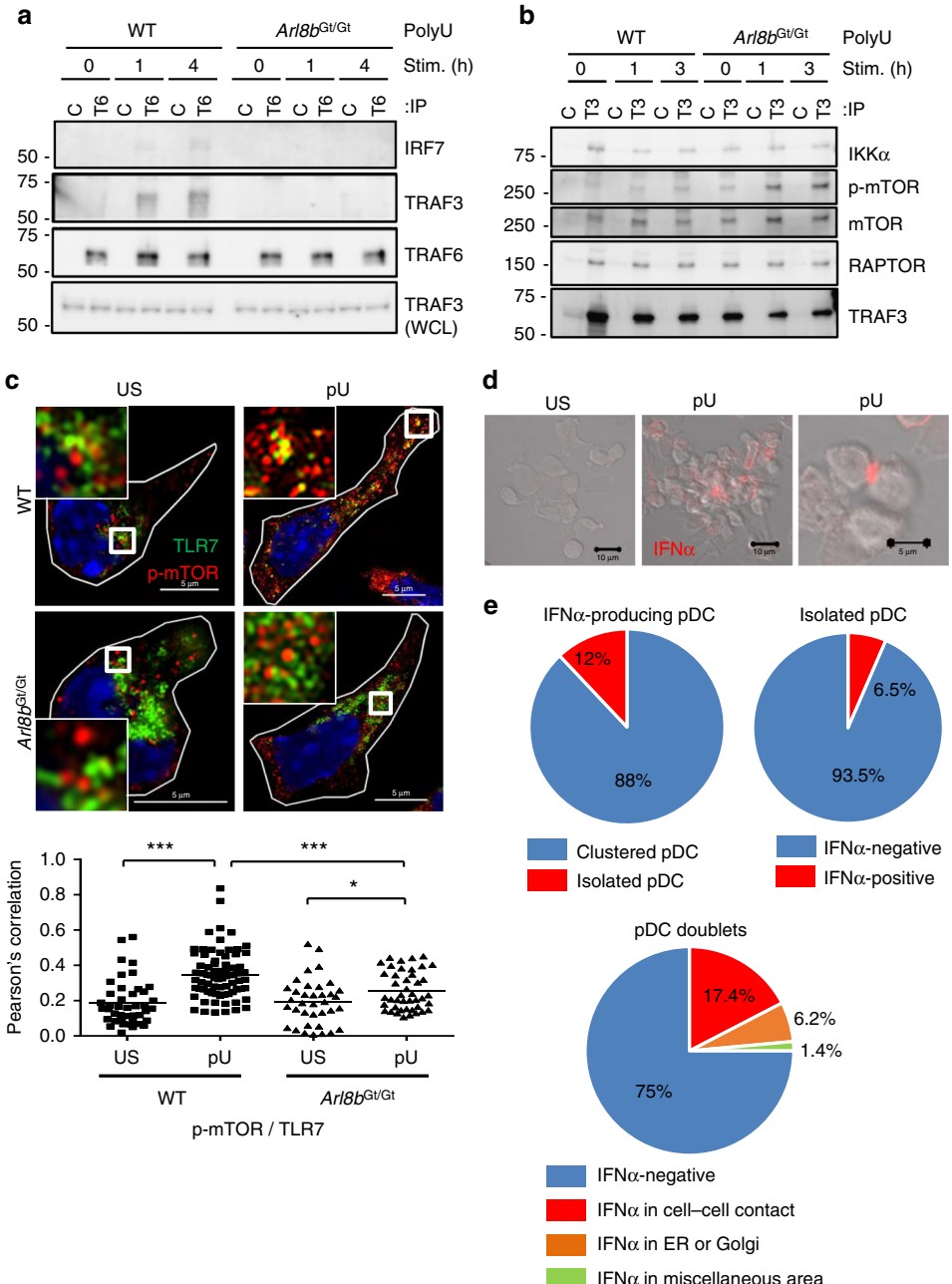

**Fig. 6** TLR7 trafficking induces TRAF6–TRAF3 association and IFN-α production in pDC clusters. **a, b** WT or *Arl8b*^Gt/Gt BM-pDCs were stimulated with 25 μg/mL polyU for the indicated times prior to lysis and immunoprecipitation using Ab against TRAF6 (T6) or goat IgG negative control (C) followed by immunostaining of the indicated signaling molecules. Apparent molecular mass is indicated (left). Immunostaining of TRAF3 in whole-cell lysate is also shown in the lowermost panel (**a**). Immunoprecipitation of the same cell lysates as in **a** but using Ab against TRAF3 (T3) or rabbit IgG negative control (C) prior to immunostaining (**b**). **c** WT and *Arl8b*^Gt/Gt BM-pDCs were either left unstimulated (US) or stimulated with 25 μg/mL polyU (pU) for 3 h. Cells were then stained with Abs against TLR7 and phosphorylated mTOR (p-mTOR) as indicated. Nuclei were stained with DAPI. Higher magnification images of boxed regions are shown in the insets. Statistical determination of TLR7 co-localization with p-mTOR is shown in the panel below (*n* > 24). ***$P < 0.001$, *$P < 0.05$ (unpaired two-tailed *t*-test). Scale bar, 5 μm. **d, e** WT BM-pDCs were either left unstimulated (US) or activated (pU) with 25 μg/mL polyU for 6–10 h. Cells were stained with an Ab against Interferon α (IFNα). Scale bar, 10 μm (left and middle), 5 μm (right). In IFN-α-producing pDCs, the percentages of clustered and isolated pDCs were determined (*n* = 324). Also, in isolated pDCs, the percentages of IFN-α-positive and -negative pDCs were determined (*n* = 600). In pDC doublets, the percentages of pDCs with IFN-α in the cell–cell contact, ER/Golgi, or miscellaneous area are shown (*n* = 276)

pDCs, suggesting constitutive association of TRAF3 with IKKα and mTORC1.

Quantification analysis of the immunoblotting data of TRAF3 immunoprecipitation (Fig. 6b) showed that the association between mTOR and TRAF3 in *Arl8b*^Gt/Gt BM-pDCs was unaltered, while the polyU-dependent phosphorylation of mTOR associated with TRAF3 increased when compared with WT BM-pDCs (Supplementary Fig. 10). Considering that the proinflammatory cytokine production in *Arl8b*^Gt/Gt BM-pDCs was higher than WT BM-pDCs (Fig. 4a), we speculate that a larger amount of proinflammatory cytokines increased phosphorylation of mTOR in *Arl8b*^Gt/Gt BM-pDCs.

The result that TRAF6 associated with TRAF3 and IRF7 in a manner dependent on TLR7 trafficking suggested that TLR7 traffics to TRAF3 and IRF7. TRAF3 was, however, found to be abundantly expressed and therefore difficult to see activation-dependent co-localization with TLR7. Instead, co-localization of TLR7 with p-mTOR, which is constitutively associated with TRAF3 was detectable (Fig. 6c). Co-localization of TLR7 and p-mTOR was increased upon polyU stimulation in WT pDCs. Increases in co-localization was much lower in $Arl8b^{Gt/Gt}$ pDCs. These results suggested that ligand-activated TLR7 traffics to the complex consisting of TRAF3, IKKα, and mTORC1 and induces association of TRAF6 with the complex to induce IFN-α.

We detected the biphasic phosphorylation of NF-κB p65 at 30–60 min and 3–4 h (Supplementary Fig. 11a, b). mTORC1 activation is required for IFN-α/β production. In the late phase of NF-κB activation, phosphorylation of mTOR and its substrates such as S6 kinase and S6 was also detected (Supplementary Fig. 11a). TLR7 was detected in the cell periphery at 3 h, but not 0.5 h, after polyU stimulation (Supplementary Fig. 11c), suggesting that TLR7 trafficking precedes the late phase of NF-κB activation. These results are consistent with the possibility that TLR7 trafficking is required for IFN-α induction by the delayed activation of NF-κB and mTORC1. These results suggest that ligand-activated TLR7 recruits TRAF6, induces proinflammatory cytokine production, and traffics to the cell periphery to enable TRAF6 interaction with TRAF3 and IRF7 (Supplementary Fig. 12).

**pDC cell adhesion restricts IFNα production to clustered pDCs.** Finally, we explored the importance of cell adhesion in IFN-α secretion. It was known that pDCs rapidly produce IFN-α/β upon viral infection[2]. Paracrine/autocrine IFN-α/β signaling is required to induce prompt and robust pDC responses[6, 7, 34]. Cell adhesion-dependent IFN-α/β production may facilitate the paracrine action of IFN-α/β. To address this possibility, we stained IFN-α in activated pDCs. IFN-α was detected preferentially in pDC clusters (Fig. 6d). Statistical analyses demonstrated that over 80% of IFN-α-producing cells were detected in pDC clusters (Fig. 6e). The percentage of IFN-α-producing cells increased from 6.5% in isolated pDCs to 25% in doublet cells. Interestingly, IFN-α was detected at cell–cell contact in 17.4% out of 25 % (Fig. 6d, e). These results strongly suggest that cell adhesion restricts IFN-α production to clustered pDCs. Directional secretion is likely to facilitate paracrine action of IFN-α, which is required for activation of maximal pDC responses[6–8].

## Discussion

pDCs rapidly produce IFN-α/β upon viral infection[2]. Paracrine/autocrine IFN-α/β signaling is required to induce maximal pDC responses[6, 7, 34,], suggesting that IFN-α/β induction is a decision to initiate full activation in pDCs. The present study suggests that IFN-α induction requires cell adhesion. The role for TLR7 trafficking in IFN-α induction is to make IFN-α induction contingent on active cell adhesion. TLR7-activated cell adhesion molecules such as CD11a/CD18 integrin, leading to cluster formation, which is formed also in vivo during virus infection[4]. pDC clusters are likely to be the principal site of IFN-α production. IFN-α production in pDC cluster would facilitate paracrine IFN-α signaling, initiating IFN-α-dependent positive feedback loop. These results suggest that cell adhesion is linked with IFN-α induction to activate paracrine IFN-α signaling in pDC clusters.

TLR7-dependent activation of CD11a/CD18 integrin induced clustering through polymerization of actin and microtubule. The inside-out signaling pathway from TLR7 to CD11a/CD18 integrin required MyD88. Our data indicate that the inside-out signaling pathway downstream of MyD88 is the PKC-PKD1-Rap1

pathway. In the case of TLR4, TLR4/MD-2-ligation by lipopolysaccharide (LPS) activates CD11b/CD18 integrin in a manner dependent on MyD88, p38, Rap1, and Ras association domain family member 5 (RAPL)[35, 36]. This related signaling pathway may also work between TLR7 and CD11a/CD18 in pDCs. The outside-in signaling from CD11a/CD18 integrin to cytoskeleton required ILK. In the target lysis by NK cells, lytic granules are polarized by CD11a/CD18 integrin. A signaling network involving ILK, Pyk2, and paxillin mediates the outside-in signaling pathway[24]. ILK was also required for microtubule polarization in activated pDCs. The signaling network downstream of CD11a/CD18 integrin may be shared between NK cells and pDCs. Our results suggest that inside-out and outside-in signaling through CD11a/CD18 integrin enables TLR7 stabilize actin and microtubule, leading to pDC polarization and clustering.

In addition to CD11a/CD18 integrin, Arl8b was also required for microtubule elongation particularly upon influenza virus Infection. Arl8b is known to link lysosomes to microtubule by interacting with SKIP and kinesin-1[30]. In addition to ATP-dependent interaction with microtubule for vesicular trafficking, the C-terminal tail of kinesin-1 interacts with microtubule in ATP-independent manner[37]. The C-terminal domain is shown to have a role in cross-link and bundle microtubule[38]. Arl8b may contribute to microtubule polarization through its interaction with kinesin-1. In contrast to influenza virus infection, polyU stimulation did induce microtubule polymerization in $Arl8b^{Gt/Gt}$ BM-pDCs. Given that influenza virus infection activates endosomal trafficking, Arl8b might be activated directly by influenza virus infection in TLR7-independent manner and enhance TLR7-dependent microtubule polymerization in pDCs.

In NK cells, Arl8b controls polarization of lytic granules to the immune synapse with target cells[39]. SKIP and kinesin are also required for the polarization likely by linking lytic granules to microtubule. Furthermore, CD11a/CD18 is one of the main cell adhesion molecule in the immune synapse. IFN-α/β induction in pDCs has a number of similarities to the target lysis by NK cells. Target lysis by NK cells is mediated by secretion of the content of lytic granules to the synapse. Given the similarity between target lysis by NK cells and IFN-α/β induction by pDCs, IFN-α may be secreted to the contact site with the adjacent cells. In contrast to IFN-α, proinflammatory cytokines do not have any link with cell adhesion. Two distinct types of cytokine production remind us cytokine secretion pathways in polarized T cells[40]. IFNγ, IL-2, and IL-10 are secreted directly into the synapse, whereas IL-4, TNFα, and chemokines such as RANTES and MIP1α show multidirectional secretion. Directional secretion of the content of lytic granules is likely to inhibit bystander cell lysis by NK cells. Similarly, cell adhesion-dependent IFN-α production may inhibit multidirectional IFN-α secretion, which may lead to IFN-α/β-dependent autoimmune diseases.

Previous studies on TLR trafficking mainly focus on AP3-dependent TLR9 trafficking between endosomes and lysosomes[13–15]. In contrast to endosomal localization of steady-state TLR9, TLR7 was detected in Arl8b+ lysosomes, suggesting that endogenous TLR7 in pDCs proceeds from endosomes to lysosomes without any ligand stimulation. Although AP3 is associated with TLR9 and regulates TLR9 trafficking to lysosomes upon stimulation[13], we could not find the ligand-dependent increase in co-localization between Arl8b and AP3. Given that ligand-activated TLR7 stayed in Arl8b+ lysosome, ligand-dependent TLR7 trafficking occurs within lysosomal compartments and unlikely depends on AP3-dependent trafficking from endosomes to lysosomes. As TLR7-dependent IFN-α/β expression is shown to depend on AP3[13], it is possible that AP3 is required for constitutive TLR7 trafficking from endosomes to lysosomes.

Ligand-dependent TLR7 trafficking occurs in lysosomal compartment. Upon ligand stimulation, anterograde TLR7 trafficking is induced. The destination of TLR trafficking was demonstrated in the present study as TRAF3 and IKKα, signaling molecules for IFN-α induction. Whereas MyD88 and TRAF6 are required for proinflammatory cytokine production and likely to be recruited to perinuclear TLR7, TRAF3, IKKα, and mTORC1 were specifically required for IFN-α/β induction and therefore unlikely recruited to perinuclear TLR7 but instead wait for TLR7 trafficking in cell periphery. We failed to detect the association of TRAF6 and TRAF3 with TLR7 in polyU-activated pDCs (Supplementary Fig. 11d). Given the trafficking-dependent TRAF6–TRAF3 association, peripheral TLR7 might be close to TRAF6 and TRAF3. The relationship between TLR7, TRAF6, and TRAF3 in pDC remains to be clarified.

Cell adhesion impacts not only cell–cell contact but also TLR7-activated IFN-α signaling in pDCs. pDCs show not only homotypic cell adhesion but also heterotypic interaction with NK cells and B cells through CD11a/CD18[5, 16]. Interestingly, steady-state TLR7 expression in B cells is dependent on IFN-α/β[41, 42]. Upon activation, splenic pDCs form clusters in the marginal zone[4]. Marginal zone B cells show higher TLR7 expression than follicular B cells[41]. It is possible that TLR7 expression in resting B cells is influenced by their interaction with pDCs. Excessive interaction of B cells and pDCs may lead to TLR7-dependent B-cell activation due to increased TLR7 expression. Such homeostatic B-cell activation may predispose to autoimmune diseases. Considering that cell adhesion molecules such as CD11a/CD18 contributes to IFN-α production by pDCs. Cell adhesion molecules would be a promising target in therapeutic intervention in IFN-α/β-dependent autoimmune diseases such as systemic lupus erythematosus.

## Methods

**Mice.** C57BL/6 mice were purchased from Japan SLC Inc. $Tlr7^{-/-}$ mice were kindly provided by Prof. S. Akira (Osaka Univ., Japan). $Itgal^{-/-}$ mice were a kind gift from Prof. Tak W. Mak (Univ. Toronto, Canada) via the RIKEN Bioresource Center (Japan). $Arl8b^{Gt/Gt}$ gene trap mice were established using ES cells (AK0793) purchased from Genome Research Limited via the Mutant Mouse Regional Resource Center (MMRRC). The mice were backcrossed onto a C57BL/6N slc strain eight times. $Plekhm2^{tm1a(EUCOMM)Wtsi}$ mice were purchased from the European Mouse Mutant Archive (EMMA). $Plekhm2^{tm1a(EUCOMM)Wtsi}$ mice were mated with CAG-cre mice to obtain $Plekhm2^{-/-}$ mice. All mice were maintained under specific pathogen-free conditions in the animal center of the Institute of Medical Science at The University of Tokyo (IMSUT). All mouse experiments were approved by the institutional animal care and use committee of IMSUT.

**Reagents.** The monoclonal Ab (1 or 2 μg/mL as final concentration) against mouse TLR7 (A94B10, mouse IgG1/κ), TLR9 (J15A7, mouse IgG1/κ), and polyclonal Abs (pAbs) against Arl8a/b were generated as described previously[28, 43, 44]. Anti-TLR7C rabbit pAb was prepared by immunizing each animal four times over a 3-month period with the GST fusion protein containing the extracellular domain of TLR7 (Alanine 583 to lysine 701). Rabbit anti-TLR7N pAb was purchased from eBioscience. Anti-Unc93b1 rabbit pAb was prepared by immunizing each animal four times over a 4-month period with GST fusion protein containing the Unc93B1 N-terminal cytoplasmic region prior to harvesting the blood sera for use in immunoblotting experiments. The chicken anti-mouse Rab7a Ab was a kind gift from Prof. Ge-Hong Sun-Wada (Doshisha Woman's College, Kyoto, Japan). The hybridoma producing rat mAb against mouse LAMP-2 (ABL-93) was obtained from the Developmental Studies Hybridoma Bank and the Ab purified prior to use. Ab obtained from commercial sources included rat anti-α-tubulin (NOVUS Biologicals), rat anti-mouse IFN-α (Hycult Biotech), rat anti-CD11a (M17/4), CD102 (3C4), CD18 (M18/2), and CD54 (YN1/1.7.4) (Biolegend), rabbit anti-Rab5, Rab7a, PKD, p-PKD (Ser916), p-PKD (Ser744/748), IKKα, MyD88, NF-κB p65, p-NF-κB p65 (Ser536), p-S6 (Ser240/244), S6 ribosomal protein, p-S6 kinase (Thr389), S6 kinase, Rap1A/Rap1B, p-mTOR, mTOR, and RAPTOR (Cell Signaling), rabbit anti-IRF7 (Bio-Rad AbD Serotec), rabbit anti-calnexin and TRAF6 (Abcam), goat anti-TRAF6 and rabbit anti-TRAF3 Abs (Santa Cruz), mouse anti-actin (Sigma-Aldrich), mouse anti-AP3 δ (DSHB). Alexa488-labeled Transferrin and goat anti-mouse IgG, DAPI staining solution, and Alexa568-labeled goat anti-rat IgG and anti-rabbit IgG were obtained from Invitrogen. Alexa488-labeled donkey anti-chicken IgY was from Jackson Immuno Research.

Alexa488-labeled rat IgG2a, IgG2b, anti-mouse CD11a, CD54 and CD102, and FITC-labeled mouse CD18 and rat IgG2a were obtained from Biolegend. ILK inhibitor (Cpd22) was obtained from Calbiochem.

Recombinant murine granulocyte-macrophage colony-stimulating factor (GM-CSF), macrophage colony-stimulating factor (M-CSF), and Fms-like tyrosine kinase-3 ligand (Flt3-L) were purchased from Peprotech. TLR ligand loxoribine was obtained from Alexis.

CpG-A 1585- (G*G*GGTCAACGTTGAG*G*G*G*G*G, asterisks indicate phosphorothioated sites) was synthesized by Hokkaido System Science. PolyU (UUUUUUUUUUUUUUUUUUUU, all phosphorothioated) was synthesized by FASMAC.

Influenza virus A/Puerto Rico/8/34 (H1N1) (PR8) was grown in the allantoic cavities of 10-day-old fertile chicken eggs at 35 °C for 2 days. Virus was stored at −80 °C and the viral titer was quantified in a standard plaque assay using MDCK cells.

**Preparation of BM-cDCs, BM macrophages, and BM-pDCs.** Bone marrow cells were cultured at 37 °C for 1 week in RPMI-1640 medium containing 10% FCS and 100 μM 2-mercaptoethanol (2ME). Cultures were supplemented with GM-CSF (10 ng/mL), M-CSF (100 ng/mL), or Flt3-L (30 ng/mL) to promote differentiation into BM-cDCs, BM macrophages, or BM-pDCs, respectively. Differentiation into BM-cDCs or BM macrophages was confirmed by staining for CD11c or CD11b. To obtain BM-pDCs, B220+ cells were sorted from total Flt3-L-treated BM cells using a FACS Aria.

**Affinity purification-mass spectrometry.** Flag-green fluorescent protein (GFP) or Flag-GFP tagged Unc93B1 were expressed in BM-cDCs. BM-cDCs were lysed in lysis buffer containing 1% (w/v) digitonin, 20 mM Tris (pH 7.4), 150 mM NaCl, 1 mM CaCl₂, 1 mM MgCl₂, 1 mM DTT, 10% Glycerol, 1x Halt Phosphatase Inhibitor Cocktail (Thermo Scientific), EDTA free complete protease inhibitor cocktail tablet (Roche). Unc93B1 and its interacting proteins were immunoprecipitated with anti-Flag mAb (M2 agarose, Sigma-Aldrich) and eluted with elution buffer containing 0.1 μg/mL Flag peptide (Sigma-Aldrich) after washing with washing buffer containing 0.1% digitonin. Flag-GFP was used as a negative control. BM macrophages were lysed in 1% (w/v) digitonin lysis buffer. TLR7 and its interacting proteins were also precipitated with anti-TLR7 Ab and eluted with SDS sample buffer (Nacalai). Isotype-matched control Ab was used as a negative control. The proteins were precipitated twice by methanol chloroform precipitation to remove the detergent and salts. The precipitated proteins were collected by centrifugation and dried completely with a centrifuge evaporator. The dried proteins were dissolved with 8 M urea, 50 mM Tris-HCl, pH 8.0, 10 mM EDTA, pH 8.0, and 0.005% n-Dodecyl-β-D-maltopyranoside (DM), and 10 mM DTT. The proteins were reduced at 37 °C for 20 min, followed by alkylation by incubation at 25 °C for 20 min in the dark with 20 mM iodoacetamide. The proteins were digested with 500 ng of trypsin (modified trypsin, Promega) at 37 °C for 12 h. The reaction was stopped by acidification with 5% formic acid to a pH lower than 2.5. Samples were desalted and concentrated by using slightly modified Stage Tips protocol. Desalted peptides were dried with a centrifuge evaporator and dissolved with 8 μL of 5% formic acid.

The LTQ-Orbitrap (Thermo Fisher Scientific) or LTQ-Orbitrap XL was equipped with an Agilent 1100 liquid chromatography system, which was modified to have a 200–300 nL/min flow rate by an in-house flow splitter. A homemade electrospray ionization tip column (100 μm internal diameter × 150 mm length) was packed with Inertsil ODS-3 C18 (3 μm, GL Sciences). The sample (4 μL) was injected to the LC-MS/MS system, and peptides were separated using a 95.5 min linear gradient of 5–28% acetonitrile in 0.125% formic acid. The LTQ-Orbitrap was operated in data-dependent acquisition mode. Full MS scans (m/z range 350–1500) were acquired with a resolution of 60,000 in the Orbitrap analyzer. The 10 most intense ions were fragmented using collision-induced dissociation and MS/MS spectra were acquired in the ion trap. All runs were performed in duplicates.

Tandem mass spectra from raw files were extracted by a software tool suite for proteomics developed in Gygi lab from Harvard Medical School and submitted to the Mascot program (Matrix Sciences) for database searching against the SwissProt mouse sequence database supplemented with protein sequences from cRAP, a database of common contaminating proteins by the Global Proteome Machine Organization (112 sequences from http://www.thegpm.org/crap/index.html) and in-house registered sequences (e.g., GFP tagged mouse Unc93B1), using the following parameters; maximum missed cleavage: 1, static modification: carbamidomethylcysteine, variable modification: methionine oxidation and serine, threonine, and tyrosine phosphorylation, mass tolerances for precursor and fragment ions of 50 ppm and 0.8 Da, respectively. Peptide- and protein-level false discovery rates were filtered to 1% using the target-decoy strategy to distinguish correct and incorrect identifications.

To determine interacting proteins of Unc93B1, we used a statistical analysis tool, Significance Analysis of INTeractome (SAINT, v 2.3.4). The peptide-spectrum match (PSM) of each protein was processed with a burn-in period of 2000, main iterations of 10,000, LowMode of 1, MinFold of 1, and Normalize of 0. PSMs of Arl8a and Arl8b were the sum of shared peptides and each unique peptide because of their highly homologous sequences. We defined the proteins with the probability score of more than 0.9 by SAINT as candidates of Unc93B1 enriched proteins.

**Real-time PCR**. Total RNA was extracted from cells with RNeasy Kit (Qiagen). About 0.5 µg of RNA was used for first-strand complimentary DNA synthesis with ReverTra Ace qPCR RT Kit (TOYOBO). 7300 Fast Real-Time PCR System (Applied Biosystems) was used for quantitative PCR assays with TaqMan Gene Expression probes. As TaqMan probes, Arl8b (Mm00482600), Arl8a (Mm01293357), TLR7 (Mm00446590), β-actin (Mm00607939), and PLEKHM2 (Mm01351044_m1) were used. The mRNA expression level of the indicated molecules was normalized by β-actin mRNA expression in each sample.

**Immunoprecipitation, pull-down, and immunoblotting**. After sorting, $2 \times 10^7$ pDCs were activated with polyU, and washed with PBS. Cells were lysed in lysis buffer consisting 0.3% CHAPS or 0.3% CHAPS, and 1% IGEPAL CA-630 (Nonidet P-40; Sigma-Aldrich), 40 mM Hepes (pH 7.4), 120 mM NaCl, 1 mM $MgCl_2$, 1 mM EDTA, 1x Halt Phosphatase Inhibitor Cocktail, EDTA free complete protease inhibitor cocktail tablet. For pull-down experiment, 2% glycerol was added. After incubation for 30 min on ice, lysates were centrifuged at 14,500 rpm for 20 min and debris was removed. The N-hydroxysuccinimide-activated Sepharose 4FF beads coupled with anti-GFP antibody, or anti-TLR7 antibody were used for immuno-precipitation. Magnetic Protein G beads, Dynabeads (Invitrogen), or FG beads (Tamagawa Seiki Co.) were also used for immunoprecipitation. For pull-down assay, RalGDS RBD beads (CELL BIOLABS, Inc) was used. Cell lysates were rotated with these beads for 2 h at 4 °C. Beads were washed with the lysis buffer once and washing buffer (0.3% CHAPS or 0.3% CHAPS and 0.5% Nonidet P-40, 40 mM HEPES (pH 7.4), 120 mM NaCl) three times. The bound proteins were subjected to SDS–PAGE. Separated proteins were transferred to polyvinylidene difluoride (PVDF) membranes and detected by immunoblotting with Can Get Signal (TOYOBO).

The AP-MS analyses are described in detail in the extended experimental procedures. Full-length uncropped blots are presented in Supplementary Figs. 13–15.

**Inhibitor experiment**. About $1 \times 10^5$ pDCs were preincubated with an ILK inhibitor (Cpd22; Calbiochem), LFA-1 inhibitor (RWJ50271; Tocris), PKD inhibitor (CRT0066101; Abcam), and PKC inhibitor (Go6983; WAKO) at the indicated concentration for 1 or 2 h. After the pretreatment, pDCs were activated with the indicated concentration of polyU for 24 h. For experiment of pDCs cluster formation, microtubule polymerization, and TLR7 trafficking, pDCs were activated for 3 or 4 h with 25 µg/mL polyU.

**Cytokine measurement by ELISA**. BM-pDCs were plated into 96-well plates (BD Falcon) at a concentration of $1 \times 10^5$ cells per well and then stimulated with TLR ligands for 24 h. Supernatant concentrations of IL-12p40 and IFN-α were determined using Ready-Set-Go ELISA kits (eBioscience, #88-7120-88) and IFNα ELISA kits (PBL Assay, #42120-2).

**Confocal microscopy and structured illumination microscopy**. FACS-sorted BM-pDCs were allowed to adhere to collagen-coated coverslips overnight before transfer into Flt3-L-deficient medium for 1 h prior to stimulation with polyU. Cells were fixed at the indicated times after polyU stimulation using 4% paraformaldehyde for 10 min, permeabilization with 0.2% Saponin in PBS for 30 min, and finally blocking with 2.5% BSA Blocking One (Nacalai) for 30 min. The cells were then incubated with primary antibodies for 90 min at 37 °C, washed, then incubated for a further for 90 min at 37 °C with secondary Abs conjugated to AlexaFluor-488 or −568 (Invitrogen). Unless specified otherwise, all microscopy was performed using a Zeiss LSM 710 apparatus and Nikon Structured illumination microscopy (N-SIM) at excitation wavelengths of 405, 488, 546, or 541 nm with a ×63 NA1.4 Plan-Apochromat oil immersion lens (Carl Zeiss Microscopy) or 100xH NA1.49 CFI Apochromat TIRF (N-SIM, Nikon). Data Acquisition was performed in 3D SIM mode before image reconstruction in NIS-Element software. To divide the perinuclear and peripheral regions of individual cells, a blue line was drawn around the nucleus and a yellow line bisecting the cell was drawn at the mid-point between the nucleus and outermost extremity of the plasma membrane. TLR7 fluorescence intensity in each compartment was then calculated in NIS-Element. Peripheral TLR7 was calculated as a percentage of total staining across both regions. Co-localization and Pearson's correlation were calculated by Velocity (PerkinElmer). Visual areas of 224.5 µm$^2$ each were used for counting of pDC clusters. Each image is representative of at least three independent experiments. Statistical significance was determined using two-sided $t$-tests.

**Data availability**. The data generated in this study are available from corresponding author upon reasonable request.

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

## Acknowledgements

We thank Profs. Tak W. Mak (Univ Toronto, Canada) and Shizuo Akira (Osaka Univ, Japan) for providing the mutant mouse strains used in this study. *Itgal*⁻/⁻ mouse strain (RBRC00850) was provided by RIKEN BRC through the National Bio-Resource Project of the MEXT, Japan. We also thank Profs. Ge-Hong Sun-Wada (Doshisha Woman's College, Japan) and Yoh Wada (Osaka Univ., Japan) for the kind gift of the anti-Rab7a antibody. We also thank Dr. Noriko Tokai, Imaging Core Laboratory in the Institute of Medical Science, The University of Tokyo, and Nikon Imaging Core Laboratory for helping us analyze the imaging results. We also thank Mr. Yuta Shimizu for technical support. This work was supported in part by Grant-in-Aid for Scientific Research (A) and (S) to K.M. (25253032, 16H06388); for Scientific Research (B) to S.-I.S. (26293083); Collaborative Research Grant from Daiichi Sankyo Co. Ltd.; the Grant for Joint Research Project of the Institute of Medical Science, the University of Tokyo; and Takeda Science Foundation. Dr. Neil McCarthy of Insight Editing London critically reviewed the manuscript.

## Author contributions

S.-I.S. conducted most experiments and wrote the manuscript. F.A., A.K., N.T., Y.M.S., R.F., T.S., K.S., T.I., M.H., Ka.Ku., H.K.-H, M.O., and Y.K. conducted part of the experiments. T.K. advised the experiments. Ke.Ko. and K.M. wrote the manuscript.

## Additional information

**Competing interests:** The authors declare no competing financial interests.

