## [Peer Review File · Nature Communications]

Reviewers' comments:

Reviewer #1 (Remarks to the Author):

In the study by Saitoh et al., authors examined a new trafficking mechanism of TLR7 endosome upon RNA stimulation. Through genetic deletion and structured illumination microscopy experiments in ex vivo cultured pDCs, authors identify that a small GTPase, Arl8b, is critical for controlling translocation of TLR7 endosome to the periphery of the cell, and formation of pDC clusters upon TLR7 stimulation by ligands. Interestingly, Arl8a/b only co-immunoprecipitated with TLR7 but not TLR9, and is only required for TLR7 but not TLR9 induction of IFNs.

This study adds valuable insights to the field of TLR trafficking and signaling. It provides the first evidence that integrins control endosomal receptor trafficking through control of microtubule polymerization and elongation. It is also the first to demonstrate that IFN is produced at the focal point between two pDCs. Finally, it provides additional insights into the differences in trafficking and signaling of TLR7 vs TLR9. Overall, it is an important study that is well written and well executed.

1. As the phenotypes of both Arl8bGt/Gt and Plekhm2^{-/-} BM-pDCs resemble those of AP-3^{-/-} BM-pDCs, AP-3 is likely to be involved in the microtubule-dependent TLR7 trafficking proposed in this paper. It would be informative if the authors could examine whether AP-3 is localized to the Arl8b complex upon TLR7 stimulation. This would place this study in the context of what is currently known in the field and extend it nicely.

2. Figure 5b seems to show increased association between p-mTOR and TRAF3 in Arl8bGt/Gt BM-pDCs stimulated with PolyU. Is this true? Perhaps quantification of the bands in 5b can help.

Reviewer #2 (Remarks to the Author):

The manuscript by Shin-Ichiroh Saitoh et al. entitled "TLR7 mediated viral recognition results in focal type I interferon secretion by dendritic cells" investigates how type I interferon secretion after TLR7 activation depends on the subcellular localization of TLR7. The authors present how after TLR7 activation cell clusters are formed in a MyD88 and CD11a dependent manner. Also, TLR7 trafficking to the cell periphery under the control of Arl8b and SKIP allows for the secretion of type I interferon. The secretion of another proinflammatory cytokine (IL-12p40) is independent of the aforementioned mechanism. The authors claim that this differential cytokine regulation is due to a change in the TLR7 complex from a TRAF6 independent complex (that induces IL-12p40) to a TRAF6 dependent one. Finally, the authors show how IFN α is produced in the cell clusters, especially in the cell-to-cell adhesions. Although the present study presents an advancement beyond what is already known, additional experiments are required to firmly support the mechanism.

1. As controls for Itgal^{-/-} and MyD88^{-/-} the authors should include the levels of TLR7, CD11a/CD18, Unc13B, MyD88, IRAK1, IRAK4, TRAF3, TRAF6 and IKK α .

2. In Figure 1, the authors show that IFN α production depends on a signaling pathway mediated by MyD88 and CD11a activation. It is known that Rap1 can activate CD11a/CD18. Madeiros et. al. (PMID: 16111639) had shown that Rap1 can be activated by PDK1. Also, Li et. al. (PMID: 25170774) had shown that PDK1 inhibition prevents type I interferon transcription downstream of TLR7/9 activation. Furthermore, Park et. al. (PMID: 19414785) showed that PDK1 activation is downstream of MyD88. Therefore it could be plausible that the MyD88, PDK1 and Rap1 axis is responsible for the effect. The authors should check the activation states of PDK1 and Rap1 after TLR7 activation at different time points and if the inhibition of those states prevents IFN α production.

3. In Figure 1A the authors should plot the SD.
4. In Figure 1B the authors should stain for tubulin for consistency.
5. To discard the option of reduced IFN α secretion due to impaired migration and/or other mechanisms instead of CD11a cell-to-cell interaction, neutralizing anti-LFA or anti-ICAM-1 and anti-ICAM-2 antibodies should be used in control pDCs. Cell clustering, tubulin polymerization, TLR7 translocation and cytokine secretion should be assessed.
6. Regarding Figure 2a, the cells that translocate TLR7 to the periphery are the ones found in clusters or are all of them?
7. As a conclusion to Figure 2a-d, the authors state that TLR7-containing lysosomes translocate from a perinuclear location to the cell periphery. Although this statement is very plausible, they do not show how lysosomes are also trafficked to the cell periphery. In supplementary figure 2, the authors should also include a quantification of lysosome and TLR7 co-localization trafficking to the periphery.
8. In supplementary Figure 2, it will also be interesting to show if there is still a high correlation between LAMP2 and TLR7 in the cell periphery to discard any effect of IFN α induction due to TLR7 translocation to another cell compartment.
9. In Figure 3, the authors show how TLR7 trafficking and IFN α production depend on Arl8a and SKIP. As controls for both knock out animal models, the authors should include the levels of TLR7 (for Plekhm2 $^{-/-}$), CD11a/CD18, Unc13B, MyD88, IRAK1, IRAK4, TRAF3 and TRAF6. Co-localization of TLR7 and LAMP2 should also be shown. Importantly, tubulin polymerization after TLR7 agonist treatment should be presented too, as it seems, from Figure 4C, that there might be a defect in tubulin polymerization in Arl8bGt/Gt BM-pDC. In case that Arl8bGt/Gt BM-pDC fail to polymerize tubulin, another approach should be used to prove the role of Arl8b in TLR7 vesicle trafficking and not tubulin polymerization.
10. Figure 3C and Figure 2D use the same graph for WT BM-pDCs, the authors should state that in the figure legend.
11. In Figure 3D different elisas for IFN α and IL-12p40 are plotted. Although they apparently use the same conditions, levels of cytokine production vary among experiments (for example, in figure 1C WT BM-pDCs after 24h of poly-U treatment at 15 μ g/ml produce 0.2ng/ml of IL-12p40 whereas in figure 3D they produce 2ng/ml). How do the authors explain these variations?
12. In Figure 3E there is a misplaced "Plekhm2 $^{-/-}$ "
13. The authors claim that "TLR7 activates cell surface integrin CD11a/CD18 and lysosomal Arl8b-SKIP to induce microtubule polymerization and co-localization of TLR7 with extended microtubules.", however, there is no evidence for the last statement.
14. In Figure 4A, the SD should be plotted for the number of cell clusters.
15. In Figure 4C, the authors show the correlation between TLR7 and α -tubulin, however they should present the percentage of TLR7 in the cell periphery because the authors claim that TLR7 localization is important for IFN α production after TLR7 activation and for consistency as well.
16. In Figure 5A and B, the authors immunoprecipitate TRAF6 or TRAF3 to show how different TLR7 complexes are formed and correlate that with TLR7 translocation to the periphery to explain the differences between proinflammatory cytokine and IFN α production. However, TRAF6 and

TRAF3 are known to interact with other proteins and the obtained results may not be describing the TLR7 complex. The authors should immunoprecipitate TLR7 at different time points and blot for TRAF3, TRAF6, IRF3, IKK α , p-mTOR, mTOR and Raptor. At the same time points, the authors should quantify the peripheral TLR7 to correlate the different complexes with TLR7 localization. The authors could also use the same time course and check by immunoblot of whole cell lysates for the activation status of NF- κ B pathway and IRF1 as both pathways are known to induce IL-12p40 as well as IRF7 for IFN α induction. These experiments will also help to better explain Figure 5C.

Point by point reply

Reviewer #1 (Remarks to the Author):

In the study by Saitoh et al., authors examined a new trafficking mechanism of TLR7 endosome upon RNA stimulation. Through genetic deletion and structured illumination microscopy experiments in ex vivo cultured pDCs, authors identify that a small GTPase, Arl8b, is critical for controlling translocation of TLR7 endosome to the periphery of the cell, and formation of pDC clusters upon TLR7 stimulation by ligands. Interestingly, Arl8a/b only co-immunoprecipitated with TLR7 but not TLR9, and is only required for TLR7 but not TLR9 induction of IFNs. This study adds valuable insights to the field of TLR trafficking and signaling. It provides the first evidence that integrins control endosomal receptor trafficking through control of microtubule polymerization and elongation. It is also the first to demonstrate that IFN is produced at the focal point between two pDCs. Finally, it provides additional insights into the differences in trafficking and signaling of TLR7 vs TLR9. Overall, it is an important study that is well written and well executed.

1. As the phenotypes of both *Arl8b*^{Gt/Gt} and *Plekhm2*^{-/-} BM-pDCs resemble those of *AP-3*^{-/-} BM-pDCs, AP-3 is likely to be involved in the microtubule-dependent TLR7 trafficking proposed in this paper. It would be informative if the authors could examine whether AP-3 is localized to the Arl8b complex upon TLR7 stimulation. This would place this study in the context of what is currently known in the field and extend it nicely.

Reply to point 1:

We could not see high co-localization between Arl8 and AP3 δ in steady-state BM-pDCs. Unexpectedly, polyU stimulation decreased the co-localization between Arl8 and AP3 δ (Supplementary Fig.8b). These results suggest that TLR7 trafficking is distinct from TLR9 trafficking, which increases co-localization between Arl8 and AP3 δ .

The 11th paragraph in Results (page 9) was newly added.

The phenotypes found in both *Arl8b*^{Gt/Gt} and *Plekhm2*^{-/-} BM-pDCs reminded us of *Ap3b1*^{-/-} BM-pDCs, in which TLR7 dependent expression of IFN-1, but not inflammatory cytokines, was impaired¹³. Ligand-dependent TLR9 trafficking to lysosomes requires adaptor protein 3 (AP3)-dependent endosomal trafficking¹³. Arl8b-dependent lysosomal trafficking studied here may also depend on AP3. The co-localization between Arl8 and AP3 δ , a subunit of AP3, was analyzed in WT BM-pDCs. We could not see high co-localization between Arl8 and AP3 δ in steady-state BM-pDCs. Unexpectedly, polyU stimulation decreased the co-localization between Arl8 and AP3 δ (Supplementary Fig. 9). These results show sharp contrast to the previous finding that co-localization between TLR9 and AP3 increased by ligand stimulation¹³.

The 5th paragraph in Discussion (page 15) was changed as below.

Previous studies on TLR trafficking mainly focus on AP3-dependent TLR9 trafficking between endosomes and lysosomes¹³⁻¹⁵. In contrast to endosomal localization of steady-state TLR9, TLR7 was detected in Arl8b⁺ lysosomes, suggesting that endogenous TLR7 in pDCs proceeds from endosomes to lysosomes without any ligand stimulation. Although AP3 is associated with TLR9 and regulates TLR9 trafficking to lysosomes upon stimulation¹³, we could not find the ligand-dependent increase in co-localization between Arl8b and AP3. Given that ligand-activated TLR7 stayed in Arl8b⁺ lysosome, ligand-dependent TLR7 trafficking occurs within lysosomal compartments and unlikely depends on AP3-dependent trafficking from endosomes to lysosomes. As TLR7-dependent IFN-1 expression is shown to depend on AP3¹³, it is possible that AP3 is required for constitutive TLR7 trafficking from endosomes to lysosomes.

2. Figure 5b seems to show increased association between p-mTOR and TRAF3 in *Arl8b*^{Gt/Gt} BM-pDCs stimulated with PolyU. Is this true? Perhaps quantification of the bands in 5b can help.

Reply to point 2:

Thank you for your constructive advice. We quantified the bands in Figure 6b by ImageJ. We could not see any increase of the association between mTor and TRAF3, while we observed the more significant increase of the association between p-mTor and TRAF3 in *Arl8b*^{Gt/Gt} BM-

pDCs stimulated with polyU compared with WT BM-pDCs. We speculate that proinflammatory cytokines from pDC at early phase of polyU activation activated the mTORC1 associated with TRAF3. New data are shown in Supplementary Fig.10

The 14th paragraph in Results (page 11) was newly added.

Quantification analysis of the immunoblotting data of TRAF3 immunoprecipitation (Fig. 6b) showed that the association between mTOR and TRAF3 in *Arl8b^{Gt/Gt}* BM-pDCs was unaltered, while the polyU-dependent phosphorylation of mTOR associated with TRAF3 increased when compared with WT BM-pDCs (Supplementary Fig. 10). Considering that the proinflammatory cytokine production in *Arl8b^{Gt/Gt}* BM-pDCs was higher than WT BM-pDCs (Fig.4a), we speculate that a larger amount of proinflammatory cytokines increased phosphorylation of mTOR in *Arl8b^{Gt/Gt}* BM-pDCs.

Reviewer #2 (Remarks to the Author):

The manuscript by Shin-Ichiroh Saitoh et al. entitled “TLR7 mediated viral recognition results in focal type I interferon secretion by dendritic cells” investigates how type I interferon secretion after TLR7 activation depends on the subcellular localization of TLR7. The authors present how after TLR7 activation cell clusters are formed in a MyD88 and CD11a dependent manner. Also, TLR7 trafficking to the cell periphery under the control of *Arl8b* and SKIP allows for the secretion of type I interferon. The secretion of another proinflammatory cytokine (IL-12p40) is independent of the aforementioned mechanism. The authors claim that this differential cytokine regulation is due to a change in the TLR7 complex from a TRAF6 independent complex (that induces IL-12p40) to a TRAF6 dependent one. Finally, the authors show how IFN α is produced in the cell clusters, especially in the cell-to-cell adhesions. Although the present study presents an advancement beyond what is already known, additional experiments are required to firmly support the mechanism.

1. As controls for *Itgal*^{-/-} and *MyD88*^{-/-} the authors should include the levels of TLR7, CD11a/CD18, Unc13B, MyD88, IRAK1, IRAK4, TRAF3, TRAF6 and IKK α .

Reply to point 1:

We confirmed the expression level of TLR7 and CD11a/CD18 by Flow Cytometry. The protein expression levels of Unc93B1, MyD88, IRAK1, IRAK4, TRAF3, TRAF6, IRF7 and IKK α in *Itgal*^{-/-} and *MyD88*^{-/-} pDC were not altered (Supplementary Fig.1a, c, d, e, f, g). New data are shown in Supplementary Fig. 1c, d, g.

The 2nd paragraph in Results (page 4) was changed as below.

In addition to cluster formation and cytoskeletal changes, IFN-1 production was also significantly impaired in *Itgal*^{-/-} pDCs whereas IL-12p40 expression was unaltered (Fig. 1e). This was the case irrespective of whether the cells were stimulated with polyU or the alternative TLR7 ligand loxoribine (Supplementary Fig. 1b). The expression of TLR7, CD54, CD102, and signaling molecules downstream of TLR7 in *Itgal*^{-/-} and *Myd88*^{-/-} BM-pDCs were not altered (Supplementary Fig. 1a, c, d, e, f, g). To further confirm the requirement of CD11a/CD18 for IFN-1 expression, we used a CD11a/CD18 inhibitor, RWJ50271, which significantly inhibited pDC clustering, microtubule polymerization, and IFN-1 expression, but not IL-12 p40 expression in polyU-activated pDCs (Supplementary Fig.2a, b, c). Together, these data indicate that TLR7-mediated activation of integrin CD11a/CD18 is required for IFN-1 induction and suggest that pDC clusters are the critical sites of IFN-1 expression.

2. In Figure 1, the authors show that IFN α production depends on a signaling pathway mediated by MyD88 and CD11a activation. It is known that Rap1 can activate CD11a/CD18. Madeiros et. al. (PMID: 16111639) had shown that Rap1 can be activated by PDK1. Also, Li et. al. (PMID: 25170774) had shown that PDK1 inhibition prevents type I interferon transcription downstream of TLR7/9 activation. Furthermore, Park et. al. (PMID: 19414785) showed that PDK1 activation is downstream of MyD88. Therefore it could be plausible that the MyD88, PDK1 and Rap1 axis is responsible for the effect. The authors should check the activation states of PDK1 and Rap1 after TLR7 activation at different time points and if the inhibition of those states prevents IFN α production.

Reply to point 2:

Thank you for this constructive suggestion. We analyzed inside-out signaling for activation of Integrin. Because PKC family is critical for integrin activation, we used a PKC inhibitor, GO6983. GO6983 significantly inhibited IFN-1 expression after polyU stimulation (Supplementary Fig.4a). PKC phosphorylates PKD1. We observed the phosphorylation of PKD1 at Serine 744/748 and Serine 916 after TLR7 activation. Furthermore, we confirmed that a PKD inhibitor suppressed IFN-1 expression, but not IL-12p40. Finally, we observed TLR7-dependent activation of Rap1, a downstream molecule of PKD1, as judged by Rap1 pulldown assay using RalGDS Rap-binding domain. These results suggest that PKC-PKD1-Rap1 pathway is essential for an inside-out signaling for LFA activation and IFN-1 expression. New data are shown in Supplementary Fig.4.

The 5th paragraph in Results (page 6) was changed as below.

We analyzed the inside-out signaling to activate CD11a/CD18 integrin. Protein kinase C (PKC) family is critical for inside-out integrin activation in T cell^{18, 19}. We used a PKC inhibitor, GO6983, and found that GO6983 significantly inhibited IFN-1 expression (Supplementary

Fig.4a). Next, we analyzed one of the PKC downstream molecule, protein kinase D1 (PKD1)²⁰. Consistent with a previous report using macrophages²¹, PKD1 was phosphorylated upon TLR7 activation in WT pDCs but not in *Myd88*^{-/-} pDCs (Supplementary Fig.4b, c). Furthermore, we confirmed that a PKD inhibitor, CRT0066101, suppressed IFN-1 expression, but much less IL-12p40 (Fig. 2d). PKD1 associates with and activates Ras-related protein 1 (Rap1), a well-known CD11a/CD18 activator^{22, 23}. We therefore examined TLR7-dependent activation of Rap1 as judged by its binding to RaIGDS Rap-binding protein. Rap1 pull-down with RaIGDS Rap-binding domain was detected 4 h after polyU stimulation (Supplementary Fig.4d). These results suggest that the TLR7-MyD88-PKC-PKD-Rap1 axis activates CD11a/CD18 integrin.

The 2nd paragraph in Discussion (page 13) was changed as below.

TLR7-dependent activation of CD11a/CD18 integrin induced clustering through polymerization of actin and microtubule. The inside-out signaling pathway from TLR7 to CD11a/CD18 integrin required MyD88. Our data indicate that the inside-out signaling pathway downstream of MyD88 is the PKC-PKD1-Rap1 pathway. In the case of TLR4, TLR4/MD-2-ligation by lipopolysaccharide (LPS) activates CD11b/CD18 integrin in a manner dependent on MyD88, p38, Rap1, and Ras association domain family member 5 (RAPL)^{35, 36}. This related signaling pathway may also work between TLR7 and CD11a/CD18 in pDCs. The outside-in signaling from CD11a/CD18 integrin to cytoskeleton required ILK. In the target lysis by natural killer (NK) cells, lytic granules are polarized by CD11a/CD18 integrin. A signaling network involving ILK, Pyk2, and paxillin mediates the outside-in signaling pathway²⁴. ILK was also required for microtubule polarization in activated pDCs. The signaling network downstream of CD11a/CD18 integrin may be shared between NK cells and pDCs. Our results suggest that inside-out and outside-in signaling through CD11a/CD18 integrin enables TLR7 stabilize actin and microtubule, leading to pDC polarization and clustering.

3. In Figure 1A the authors should plot the SD.

Reply to point 3:

We showed the total number of cluster observed in 40 areas in 4 experiments in Fig. 1a. SD could not be calculated for this analysis.

4. In Figure 1B the authors should stain for tubulin for consistency.

Reply to point 4:

Thank you for this comment. The tubulin polymerization data are shown in Fig. 1c, d.

The 1st paragraph in Results (page 4) was changed as below.

The present study addressed the role of cell adhesion in IFN-1 induction. Cell adhesion via CD11a/CD18 integrin enhances IFN-1 production by pDCs^{5, 16}. CD11a/CD18 and its ligands, CD54 (also known as ICAM-1) and CD102 (ICAM-2), were expressed on pDCs (Supplementary Fig. 1a). To study the role of CD11a/CD18 integrin in TLR7-induced anti-viral responses, wild-type (WT) pDCs or those deficient in TLR7 (*Tlr7*^{-/-}), CD11a (*Itgal*^{-/-}), or MyD88 (*Myd88*^{-/-}) were stimulated with polyU single-stranded RNA. We found that polyU exposure induced pronounced clustering and polymerization of actin and microtubule in WT BM-pDCs but not their *Tlr7*^{-/-}, *Itgal*^{-/-}, or *Myd88*^{-/-} counterparts (Fig. 1a-d). This indicated that TLR7 activates CD11a/CD18-mediated cell adhesion through the Myd88-dependent signaling pathway.

5. To discard the option of reduced IFN α secretion due to impaired migration and/or other mechanisms instead of CD11a cell-to-cell interaction, neutralizing anti-LFA or anti-ICAM-1 and anti-ICAM-2 antibodies should be used in control pDCs. Cell clustering, tubulin polymerization, TLR7 translocation and cytokine secretion should be assessed.

Reply to point 5:

We analyzed IFN α secretion from pDC after polyU stimulation with anti-LFA1, anti-ICAM1 and anti-ICAM2 antibodies. However, these antibodies did not inhibit pDC clustering or IFN α secretion from pDCs. We speculated that these antibodies failed to inhibit LFA-1 dependent cell adhesion between pDCs. We found that an LFA-1 inhibitor, RWJ 50271, could inhibit LFA-1 dependent pDC clustering (Supplementary Fig.2a). We therefore used this inhibitor instead of those antibodies for requested experiment. RWJ 50271 significantly inhibited cell clustering, tubulin polymerization, TLR7 translocation and IFN α secretion from pDCs (Supplementary Fig.2a-d).

The 2nd paragraph in Results (page 4) was changed as below.

In addition to cluster formation and cytoskeletal changes, IFN-1 production was also significantly impaired in *Itgal*^{-/-} pDCs whereas IL-12p40 expression was unaltered (Fig. 1e). This was the case irrespective of whether the cells were stimulated with polyU or the alternative TLR7 ligand loxoribine (Supplementary Fig. 1b). The expression of TLR7, CD54, CD102, and signaling molecules downstream of TLR7 in *Itgal*^{-/-} and *Myd88*^{-/-} BM-pDCs were not altered (Supplementary Fig. 1a, c, d, e, f, g). To further confirm the requirement of CD11a/CD18 for IFN-1 expression, we used an LFA-1 inhibitor, RWJ50271, which significantly inhibited pDC clustering, microtubule polymerization, and IFN-1 expression, but not IL-12 p40 expression in polyU-activated pDCs (Supplementary Fig.2a, b, c). Together, these data indicate that TLR7-mediated activation of integrin CD11a/CD18 is required for IFN-1 induction and suggest that pDC clusters are the critical sites of IFN-1 expression.

The 4th paragraph in Results (page 6) was changed as below.

Itgal^{-/-} pDCs exhibited only weak microtubule polymerization compared with WT pDCs (Fig. 1c), while ligand-activated TLR7 displayed reduced co-localization with microtubules and failed to traffic to the cell periphery (Fig. 2a-c). **RWJ50271, the LFA-1 inhibitor, also inhibited TLR7 trafficking to the cell periphery (Supplementary Fig.2d).** These results suggest that ligand-activated TLR7 stimulates microtubule polymerization and elongation in a CD11a/CD18-dependent manner to allow translocation of TLR7-containing lysosomes from a perinuclear location to the cell periphery.

6. Regarding Figure 2a, the cells that translocate TLR7 to the periphery are the ones found in clusters or are all of them?

Reply to point 6:

The cells which translocate TLR7 to the periphery are found both inside and outside of clusters. Fig. 2a shows pDCs outside of clusters.

7. As a conclusion to Figure 2a-d, the authors state that TLR7-containing lysosomes translocate from a perinuclear location to the cell periphery. Although this statement is very plausible, they do not show how lysosomes are also trafficked to the cell periphery. In supplementary figure 2, the authors should also include a quantification of lysosome and TLR7 co-localization trafficking to the periphery.

8. In supplementary Figure 2, it will also be interesting to show if there is still a high correlation between LAMP2 and TLR7 in the cell periphery to discard any effect of IFN α induction due to TLR7 translocation to another cell compartment.

Reply to point 7 and 8:

We analyzed the co-localization of lysosome marker LAMP-2 with TLR7 in perinuclear or peripheral region. We did not see any difference in co-localization between perinuclear and peripheral regions. The data are shown in supplementary Fig. 3d.

The 3rd paragraph in Results (page 5) was changed as below.

TLR7 is localized in endosomes/lysosomes and ligand-activated trafficking of TLR7 is essential for induction of IFN-1^{13, 14, 17}. The changes in adhesive properties could potentially alter IFN-1 induction by impacting TLR7 distribution within these cells. To address this possibility, we visualized TLR7 using immunostaining and analysis via structured illumination microscopy (SIM). Both before and after polyU stimulation, TLR7 was observed to co-localize with the lysosomal marker LAMP-2 (Supplementary Fig. 3a, b) but not with early endosomal marker Rab5 (Supplementary Fig.3b, c), indicating that TLR7 is selectively localized to LAMP-2⁺ lysosomes. We next assessed whether this selective distribution of ligand-activated TLR7 depended on microtubules, which are thought to be essential for lysosome trafficking. Exposure to TLR7 ligand polyU induced microtubule polymerization and extension from the perinuclear region to the cell periphery in WT pDCs, but this process was significantly impaired in *Tlr7*^{-/-} and *Itgal*^{-/-} pDCs (Fig. 1c, 2a). TLR7 displayed significantly higher co-localization with microtubules after ligand activation, and polyU exposure increased the proportion of TLR7 localized to the cell periphery by >30% in WT pDCs (Fig. 2b, c). **The co-localization between TLR7 and LAMP-2 did not change between perinuclear and peripheral regions (Supplementary Fig. 3d), suggesting that TLR7-containing LAMP-2⁺ lysosomes translocate from perinuclear to peripheral regions.**

9. In Figure 3, the authors show how TLR7 trafficking and IFN α production depend on Arl8a and SKIP. As controls for both knock out animal models, the authors should include the levels of TLR7 (for *Plekhm2*^{-/-}), CD11a/CD18, *Unc13B*, *MyD88*, *IRAK1*, *IRAK4*, *TRAF3* and *TRAF6*. Co-localization of TLR7 and LAMP2 should also be shown. Importantly, tubulin polymerization after TLR7 agonist treatment should be presented too, as it seems, from Figure 4C, that there might be a defect in tubulin polymerization in *Arl8b*^{Gt/Gt} BM-pDC. In case that *Arl8b*^{Gt/Gt} BM-pDC fail to polymerize tubulin, another approach should be used to prove the role of *Arl8b* in TLR7 vesicle trafficking and not tubulin polymerization.

Reply to point 9:

We confirmed that TLR7 expression level was not altered in *Plekhm2*^{-/-} pDCs (Supplementary Fig. 8c). The expression of CD11a and CD18 on *Plekhm2*^{-/-} and *Arl8b*^{Gt/Gt} pDCs was also unaltered (Supplementary Fig. 6i and 8d). The protein expression of *Unc93B1*, *MyD88*, *IRAK1*, *IRAK4*, *IKK α* , *TRAF3*, *TRAF6* and *IRF7* in *Plekhm2*^{-/-} and *Arl8b*^{Gt/Gt} pDCs was comparable to WT pDCs (Supplementary Fig. 8e). LAMP-2 and TLR7 were co-localized in *Plekhm2*^{-/-} and *Arl8b*^{Gt/Gt} as much as in WT pDCs (Supplementary Fig. 8b). We observed the polyU-dependent microtubule elongation in *Arl8b*^{Gt/Gt} pDC (Supplementary Fig. 7b).

The 8th paragraph in Results (page 8) was changed as below.

Since *Arl8a* and *Arl8b* are highly homologous proteins that exhibit 91% amino acid identity (Supplementary Fig. 5c), we focused subsequent analyses on *Arl8b* which has a well-documented role in anterograde lysosomal trafficking²⁷⁻²⁹. To explore the importance of *Arl8b* in regulating TLR7 responses, we obtained *Arl8b*^{Gt/Gt} gene trap mice (Supplementary Fig. 6a) and assessed receptor distribution and function in pDCs. While a considerable number of *Arl8b*^{Gt/Gt} mice died from unknown causes, the rest of animals developed normally (Supplementary Fig. 6b, c). *Arl8b* mRNA and protein were not detectable in *Arl8b*^{Gt/Gt} pDCs, whereas the levels of *Arl8a* mRNA and protein were unaltered (Supplementary Fig. 6d, e). The expression levels of TLR7 mRNA and protein were also comparable with WT controls (Supplementary Fig. 6f,

g, h). We then assessed the impact of *Arl8b* deficiency on polyU-activated microtubule polymerization and TLR7 trafficking in pDCs. PolyU-induced pDC clustering was comparably observed in WT and *Arl8b^{Gt/Gt}* cells (Supplementary Fig. 7a). Also, polyU induced microtubule polymerization in *Arl8b^{Gt/Gt}* BM-pDCs (Supplementary Fig. 7b). However, ligand-activated TLR7 failed to co-localize with microtubules and to traffic to the cell periphery in *Arl8b^{Gt/Gt}* BM-pDCs (Fig. 3c, d). These results suggest that *Arl8b* is required for the interaction between TLR7-containing lysosomes and microtubule but not receptor-mediated activation of integrin CD11a/CD18.

The 9th paragraph in Results (page 8) was changed as below.

Arl8b binds to SifA and kinesin-interacting protein SKIP (also known as *Plekhm2*), which contributes to the regulation of anterograde lysosomal trafficking³⁰. We therefore investigated whether SKIP influences TLR7 trafficking in pDCs by assessing receptor distribution in *Plekhm2^{-/-}* pDCs, in which the lack of *Plekhm2* mRNA was verified by PCR (Supplementary Fig. 8a). Whereas polyU-activated microtubule polymerization and co-localization between LAMP-2 and TLR7 were not impaired (Supplementary Fig. 7b, 8b), the co-localization between TLR7 and microtubule and TLR7 trafficking to cell periphery were impaired in *Plekhm2^{-/-}* BM-pDCs (Fig. 3c, d). The expression of TLR7, CD11a/CD18, and TLR7 signaling molecules were not altered in *Plekhm2^{-/-}* BM-pDCs (Supplementary Fig. 8c, d, e). As expected, *Plekhm2^{-/-}* pDCs resemble *Arl8b^{Gt/Gt}* pDCs, indicating that the *Arl8b*-SKIP axis links TLR7-containing lysosomes to polymerized microtubules for the transport to the cell periphery.

10. Figure 3C and Figure 2D use the same graph for WT BM-pDCs, the authors should state that in the figure legend.

Reply to point 10:

Thank you for the comment. We changed the sentences in the legend to Figure 3 as suggested.

The legend to Figure3 is changed as below.

Figure 3. Impaired TLR7 trafficking in *Arl8b^{Gt/Gt}* and *Plekhm2^{-/-}* pDCs.

(a) TLR7 and TLR9 were immunoprecipitated from BM-pDCs and subjected to immunostaining of TLR7, TLR9, Unc93b1, and *Arl8a/b*. Lanes 1 and 3 show immunoprecipitations with isotype-matched control Ab. The lowermost blot shows immunostaining of whole cell lysate with Ab against *Arl8a/b*. Apparent molecular mass is indicated (left). (b) BM-pDCs were either left unstimulated (US) or activated (pU) with 25 µg/ml polyU for 3h prior to staining with Ab against TLR7 (green), *Arl8a/b* (red, upper panels), and Rab7a (red, lower panels). Nuclei were visualized by DAPI staining (blue). Higher magnification images of boxed regions are shown in the insets. Also shown is the statistical analysis of TLR7 co-localization with *Arl8* or Rab7a. (c) WT, *Arl8b^{Gt/Gt}*, and *Plekhm2^{-/-}* BM-pDCs were either left unstimulated (US) or activated with 25µg/ml polyU (pU) for 4h prior to Ab staining of TLR7 (green), α -tubulin (red), and DAPI staining of cell nuclei (blue). Also shown is the statistical analysis of TLR7 co-localization with microtubule. (d) WT, *Arl8b^{Gt/Gt}*, and *Plekhm2^{-/-}* BM-pDCs were either left unstimulated (US) or activated with 25µg/ml polyU (pU) for 3h prior to Ab staining of TLR7 and DAPI staining. Right panel (d) shows the percentages of peripheral TLR7. WT samples shown here are in part from Fig.2c.

11. In Figure 3D different elisas for IFN α and IL-12p40 are plotted. Although they apparently use the same conditions, levels of cytokine production vary among experiments (for example, in figure 1C WT BM-pDCs after 24h of poly-U treatment at 15µg/ml produce 0.2ng/ml of IL-12p40 whereas in figure 3D they produce 2ng/ml). How do the authors explain these variations?

Reply to point 11:

Unfortunately, the lot of polyU used in all the figures except Fig. 4a was very weak in TLR7 activation.

12. In Figure 3E there is a misplaced “*Plekhm2^{-/-}*”

Reply to point 12:

Thank you for the comment. We deleted it.

13. The authors claim that “TLR7 activates cell surface integrin CD11a/CD18 and lysosomal *Arl8b*-SKIP to induce microtubule polymerization and co-localization of TLR7 with extended microtubules.”, however, there is no evidence for the last statement.

Reply to point 13:

Thank you for the comment. We validated that polyU stimulation significantly increased the microtubule polymerization in *Plekhm2^{-/-}* and *Arl8b^{Gt/Gt}* pDCs as much as in WT pDCs. However, polyU-dependent increase in the co-localization between TLR7 and microtubules in *Plekhm2^{-/-}* and *Arl8b^{Gt/Gt}* pDCs was impaired (Fig. 3c). These results indicate that *Arl8b* and *Plekhm2* are required for TLR7 trafficking on extended microtubules. According to new findings, we changed the claim as below.

In 10th paragraph in Results (page 9) was changed as below.

Upon stimulation with polyU or loxoribine, *Arl8b^{Gt/Gt}* pDCs displayed significant impairment of IFN-1 induction (Fig. 4a and Supplementary Fig. 8f), whereas expression levels of pro-inflammatory cytokine IL-12p40 and TLR9-mediated responses to CpG-A were unimpaired. In fact, IL-12p40 production was significantly enhanced in *Arl8b^{Gt/Gt}* pDCs for a currently unknown reason. *Plekhm2^{-/-}* pDCs also displayed a selective defect in TLR7-dependent IFN-1 production comparable to that observed in their *Arl8b^{Gt/Gt}* counterparts (Fig. 4b and Supplementary Fig. 8g).

These results demonstrated that the anterograde lysosomal movement under the control of Arl8b and SKIP is required for IFN-1 induction by TLR7. Taken together with our earlier observation that TLR7 trafficking is disrupted in *Itgal*^{-/-} pDCs (Fig. 2c), these findings indicated that **TLR7 activates cell surface integrin CD11a/CD18 to induce microtubule polymerization, and the lysosomal Arl8b-SKIP axis to link TLR7 with extended microtubules.** TLR7 trafficking to the cell periphery ultimately enabled IFN-1 induction.

14. In Figure 4A, the SD should be plotted for the number of cell clusters.

Reply to point 14

We showed the total number of cluster in 30 area in 3 experiments in Fig. 5a. SD could not be calculated for this analysis.

15. In Figure 4C, the authors show the correlation between TLR7 and α -tubulin, however they should present the percentage of TLR7 in the cell periphery because the authors claim that TLR7 localization is important for IFN α production after TLR7 activation and for consistency as well.

Reply to point 15

Thank you for this comment. New data in Fig. 5d show that TLR7 localized in the cell periphery after Influenza virus infection in WT pDCs, but not in *Arl8b*^{Gt/Gt} pDCs.

The 12th paragraph in Results (page 10) was changed as below.

We next investigated the role of Arl8b in TLR7 responses to influenza virus infection of pDCs. Influenza virus induced pronounced clustering of WT pDCs and *Arl8b*^{Gt/Gt} pDCs, but not *Tlr7*^{-/-} pDCs (Fig. 5a). While virus exposure stimulated marked expression of IL-12p40 in *Arl8b*^{Gt/Gt} pDCs, these cells failed to produce IFN-1 under the same conditions (Fig. 5b). Influenza virus induced TLR7 co-localization with microtubules and extended these to the cell periphery in WT pDCs but not *Arl8b*^{Gt/Gt} pDCs (Fig. 5c). **Consequently, TLR7 was localized to the cell periphery in WT pDCs, but much less in *Arl8b*^{Gt/Gt} pDCs (Fig. 5d).** Despite exhibiting normal virus-induced clustering behavior, microtubule elongation was impaired in *Arl8b*^{Gt/Gt} pDCs. Although microtubule polymerization was detected in polyU-activated *Arl8b*^{Gt/Gt} pDCs (Supplementary Fig. 7b), we cannot exclude a possibility that Arl8b contributes partially to microtubule elongation particularly in influenza infection. These results demonstrated that, as with polyU exposure, Arl8b-dependent TLR7 trafficking in pDCs is required for IFN-1 responses to live influenza virus.

16. In Figure 5A and B, the authors immunoprecipitate TRAF6 or TRAF3 to show how different TLR7 complexes are formed and correlate that with TLR7 translocation to the periphery to explain the differences between proinflammatory cytokine and IFN α production. However, TRAF6 and TRAF3 are known to interact with other proteins and the obtained results may not be describing the TLR7 complex. The authors should immunoprecipitate TLR7 at different time points and blot for TRAF3, TRAF6, IRF3, IKK α , p-mTOR, mTOR and Raptor. At the same time points, the authors should quantify the peripheral TLR7 to correlate the different complexes with TLR7 localization. The authors could also use the same time course and check by immunoblot of whole cell lysates for the activation status of NF- κ B pathway and IRF1 as both pathways are known to induce IL-12p40 as well as IRF7 for IFN α induction. These experiments will also help to better explain Figure 5C.

Reply to point 16

We immunoprecipitated TLR7 at different time points after polyU stimulation. However, we could not see the co-precipitation of TRAF3, TRAF6, IRF7, IKK α , p-mTOR, and mTOR. Instead, we observed polyU-induced NF- κ B activation and IRF7 activation. Although, biphasic phosphorylation of NF- κ B p65 was detected at 30-60min and 3-4h (Supplementary Fig.11a, b), we could not detect polyU-induced phosphorylation of IRF7 and IKK α . To show activation of the signaling pathway for IFN-1 induction, we examined polyU-induced mTORC1 activation. The polyU-induced mTORC1 phosphorylation was detected from 2h after stimulation (Supplementary Fig.11a). To analyze the relationship between polyU-induced TLR7 signaling and trafficking, we quantified the peripheral TLR7 at different time points, 30 min and 3 hours after stimulation. We could not observe polyU-induced TLR7 trafficking to the cell periphery at 30min after stimulation, when the early phase of NF- κ B activation occurred. On the other hand, the polyU-induced TLR7 trafficking to cell periphery was observed at 3 hours after stimulation, when delayed activation of NF- κ B and mTORC1 was detected (Supplementary Fig.11c). These results suggest that TLR7 localization in the cell periphery correlates with activation of the downstream signaling for IFN α production. New data are shown in Supplementary Fig.11.

The 16th paragraph in Results (page 12) was changed as below.

We detected the biphasic phosphorylation of NF- κ B p65 at 30-60min and 3h-4h (Supplementary Fig.11a, b). mTORC1 activation is required for IFN-1 production. In the late phase of NF- κ B activation, phosphorylation of mTOR and its substrates such as S6 kinase and S6 was also detected (Supplementary Fig.11a). TLR7 was detected in the cell periphery at 3 h, but not 0.5 h, after polyU stimulation (Supplementary Fig. 11c), suggesting that TLR7 trafficking precedes the late phase of NF- κ B activation. These results are consistent with the possibility that TLR7 trafficking is required for IFN-1 induction by the delayed activation of NF- κ B and mTORC1. These results suggest that ligand-activated TLR7 recruits TRAF6, induces pro-inflammatory cytokine production, and traffics to the cell periphery to enable TRAF6 interaction with TRAF3 and IRF7.

Reviewers' comments:

Reviewer #1 (Remarks to the Author):

The authors have adequately addressed my critiques.

Reviewer #2 (Remarks to the Author):

The authors have revised the manuscript and most the questions have been addressed properly. However, some answers remain to be properly supported by data. Specifically:

-In my previous review, I asked the authors to use neutralizing antibodies anti-LFA-1 and anti-ICAM1/2. The authors claim that those antibodies did not have an effect and speculate that it could be because they failed to inhibit LFA-1 dependent cell adhesion between pDCs. This data should have been shown accompanied by the antibody clone name as a reference.

-In the final point, I asked the authors to immunoprecipitate TLR7 at different time points. The authors claim that they could not co-precipitate other known proteins that interact with TLR7. Also, they claim that IRF7 is activated after poly-U treatment although they also say that they could not detect polyU-induced p-IRF7. Therefore, it is not clear whether IRF7 is activated, and it is also not clear which different complexes are formed downstream of TLR7. As I previously stated, TRAF6 and TRAF3 do not exclusively bind to it. Moreover, those changes that the authors claim to occur to the TLR7 complex happen at latter time points when other pro-inflammatory cytokines are already produced and could be inducing an autocrine signaling.

Other than the two critiques mentioned above, the revised manuscript has been improved and no other major deficiencies have been found. Overall, these results are interesting and novel and provide new understanding of TLR trafficking and IFN α production.

Point by point reply

Reviewer #1 (Remarks to the Author):

The authors have adequately addressed my critiques.

Thank you very much for your high evaluation.

Reviewer #2 (Remarks to the Author):

The authors have revised the manuscript and most the questions have been addressed properly. However, some answers remain to be properly supported by data. Specifically:

-In my previous review, I asked the authors to use neutralizing antibodies anti-LFA-1 and anti-ICAM1/2. The authors claim that those antibodies did not have an effect and speculate that it could be because they failed to inhibit LFA-1 dependent cell adhesion between pDCs. This data should have been shown accompanied by the antibody clone name as a reference.

Thank you very much for your advice. We changed the manuscript.

Reply to point 1:

We analyzed the effect of antibodies, anti-LFA-1 and anti-ICAM1/2, on IFN-1 expression in

polyU-stimulated Bone Marrow pDCs (BMpDCs). A result is shown below. We used anti-LFA-1(BioLegend, clone M17/4), anti-ICAM-1(BioLegend, clone YN1/1.7.4), and anti-ICAM-2 (eBioscience, clone 3C4) for the experiment. According to manufacturer's instructions, these antibodies should be able to block the interaction. BMpDCs were pretreated with antibodies for 1h and stimulated for 24h with 20 $\mu\text{g/ml}$ polyU. These anti-LFA-1 and ICAM-1 antibodies did not inhibit IFN-1 production in polyU-stimulated pDC. Only anti-ICAM2 antibody slightly inhibited the IFN-1 production. Unexpectedly, these inhibitory antibodies enhanced homotypic cell adhesion. We do not know the accurate reason for this unexpected result. Instead of the antibodies, we show the results with the LFA-1 inhibitor.

-In the final point, I asked the authors to immunoprecipitate TLR7 at different time points. The authors claim that they could not co-precipitate other known proteins that interact with TLR7. Also, they claim that IRF7 is activated after poly-U treatment although they also say that they could not detect polyU-induced p-IRF7. Therefore, it is not clear whether IRF7 is activated, and it is also not clear which different complexes are formed downstream of TLR7. As I previously stated, TRAF6 and TRAF3 do not exclusively bind to it. Moreover, those changes that the authors claim to occur to the TLR7 complex happen at latter time points when other pro-inflammatory cytokines are already produced and could be inducing an autocrine signaling.

Other than the two critiques mentioned above, the revised manuscript has been improved and no other major deficiencies have been found. Overall, these results are interesting and novel and provide new understanding of TLR trafficking and IFN α production.

Reply to point 2:

We tried to prove the association of TLR7 with TRAF6 and TRAF3. BMpDCs were stimulated with TLR7 ligands polyU or R848 at different time points. The pDCs were lysed with lysis buffer including 0.5% Chaps and TLR7 was immunoprecipitated. The results are shown below. We could not observe the association of TRAF6 or TRAF3 with TLR7. Whole cell lysates (WCL) of 2.5×10^5 or 3.5×10^5 were applied for the positive control for immunoblotting of TRAF3, TRAF6 and TLR7. We also analyzed the association between TLR7 and TRAF3 or TRAF6 by using macrophage cell line, J774. However, we could not observe their association. The results are shown below.

As we could not show the evidence of TLR7-TRAF6 or TLR7-TRAF3 association, we changed the sentences in our manuscript as follows.

The 6th paragraph in Discussion (page 15 to 16) was changed as below.

Before the change

Ligand dependent TLR7 trafficking occurs in lysosomal compartment. Upon ligand stimulation, anterograde TLR7 trafficking is induced. The destination of TLR trafficking was demonstrated in the present study as TRAF3 and IKK α , signaling molecules for IFN-1

induction. Whereas MyD88 and TRAF6 are required for proinflammatory cytokine production and likely to be recruited to perinuclear TLR7, TRAF3, IKK α , and mTORC1 were specifically required for IFN-1 induction and therefore unlikely recruited to perinuclear TLR7 but instead wait for TLR7 trafficking in cell periphery. Trafficking-dependent TRAF6-TRAF3 association suggests the interaction of the two complexes consisting of TLR7-MyD88-TRAF6 and of TRAF3, IKK α , and mTORC1. The previous study shows that TLR9 trafficking enhances TRAF3 association with IRF7¹³. IRF7 is likely to be another component in the complex consisting of TRAF3, IKK α , and mTORC1. Our results suggest that TRAF3, IKK α , and mTORC1 distribute differently from MyD88 and TRAF6 in pDCs.

Revised

Ligand dependent TLR7 trafficking occurs in lysosomal compartment. Upon ligand stimulation, anterograde TLR7 trafficking is induced. The destination of TLR trafficking was demonstrated in the present study as TRAF3 and IKK α , signaling molecules for IFN-1 induction. Whereas MyD88 and TRAF6 are required for proinflammatory cytokine production and likely to be recruited to perinuclear TLR7, TRAF3, IKK α , and mTORC1 were specifically required for IFN-1 induction and therefore unlikely recruited to perinuclear TLR7 but instead wait for TLR7 trafficking in cell periphery. **We failed to detect the association of TRAF6 and TRAF3 with TLR7 in polyU-activated pDCs (data no shown). Given the trafficking-dependent TRAF6-TRAF3 association, peripheral TLR7 might be close to TRAF6 and TRAF3. The relationship between TLR7, TRAF6 and TRAF3 in pDC remains to be clarified.**~~The previous study shows that TLR9 trafficking enhances TRAF3 association with IRF7¹³. IRF7 is likely to be another component in the complex consisting of TRAF3, IKK α , and mTORC1. Our results suggest that TRAF3, IKK α , and mTORC1 distribute differently from MyD88 and TRAF6 in steady state pDCs.~~

The legend to Supplementary Figure12 (page 55) was changed as below.

Before the change

In resting cells, TLR7 is localized to lysosomes around the nuclei (N). TLR7 ligation induces production of proinflammatory cytokines and promotes cell adhesion through MyD88-dependent signaling pathways (upper panel). Activated CD11a/CD18 induces microtubule elongation through the ILK-dependent signaling pathway (lower panel). TLR7 also activates GTPase Arl8b to link TLR7-containing lysosomes with polymerized microtubule and traffic to cell periphery. Then, TLR7 interacts with the molecular complex consisting of TRAF3, IKK α , and mTORC1 in the peripheral region to induce IFN-1. IFN-1 is produced predominantly in pDC clusters.

Revised

In resting cells, TLR7 is localized to lysosomes around the nuclei (N). TLR7 ligation induces production of proinflammatory cytokines and promotes cell adhesion through MyD88-dependent signaling pathways (upper panel). Activated CD11a/CD18 induces microtubule elongation through the ILK-dependent signaling pathway (lower panel). TLR7 also activates GTPase Arl8b to link TLR7-containing lysosomes with polymerized microtubule and traffic to cell periphery. Then, TLR7 **activates** the molecular complex consisting of TRAF3, IKK α , and mTORC1 in the peripheral region to induce IFN-1. IFN-1 is produced predominantly in pDC clusters.

REVIEWERS' COMMENTS:

Reviewer #2 (Remarks to the Author):

The authors have properly addressed the different points. Although the authors were not able to co-IP TRAF6 or TRAF3 with TLR7 they have modified the text according to the observed results. The lack of this result does not affect the validity and the overall novelty of the manuscript. I recommend this work for publication.

Point by point reply

Reviewer #2 (Remarks to the Author):

The authors have properly addressed the different points. Although the authors were not able to co-IP TRAF6 or TRAF3 with TLR7 they have modified the text according to the observed results. The lack of this result does not affect the validity and the overall novelty of the manuscript. I recommend this work for publication.

Thank you very much for your high evaluation.